

# Quantification and evaluation of atmospheric pollutant emissions from open biomass burning with multiple methods: A case study for Yangtze River Delta region, China

Yang Yang[1] and Yu Zhao[1,2*]

1. State Key Laboratory of Pollution Control & Resource Reuse and School of the Environment, Nanjing University, 163 Xianlin Ave., Nanjing, Jiangsu 210023, China

2. Jiangsu Collaborative Innovation Center of Atmospheric Environment and Equipment Technology (CICAEET), Nanjing University of Information Science & Technology, Jiangsu 210044, China

*Corresponding author: Yu Zhao

Phone: 86-25-89680650; email: yuzhao@nju.edu.cn



18        **Abstract**

19        Air pollutant emissions from open biomass burning (OBB) in Yangtze River

Delta (YRD) were estimated for 2005-2015 using three (traditional bottom-up, fire
radiative power (FRP)-based, and constraining) approaches, and the differences
between those methods and their underlying reasons were analyzed. The species
included $PM_{10}$, $PM_{2.5}$, organic carbon (OC), black carbon (BC), $CH_4$, non-methane
volatile organic compounds (NMVOCs), CO, $CO_2$, $NO_X$, $SO_2$ and $NH_3$. The
inter-annual trends in emissions with FRP-based and constraining methods were
similar with the fire counts in 2005-2012, while that with traditional method was not.
For most years, emissions of all species estimated with constraining method were
smaller than those with traditional method except for NMVOCs, while they were
larger than those with FRP-based except for EC, $CH_4$ and $NH_3$. Such discrepancies
result mainly from different masses of crop residues burned in the field (CRBF)
estimated in the three methods. Chemistry transport modeling (CTM) was applied to
test the three OBB inventories. The simulated $PM_{10}$ concentrations with the
constrained emissions were closest to available observations, implying that the
constraining method provided the best emission estimates. To further evaluate the
effects of method and data on OBB emission estimation, CO emissions in this study
were compared with other national and global inventories. In general, inventories of
FRP/BA-based method might underestimate the emissions, attributed to the detection
limit on small fires. In contrast, the method based on the assumed/surveyed fraction of
burned biomass could often overestimate OBB emissions and could hardly track their
inter-annual trends. In particular, the constrained emissions in this work were close to
GFEDv4.1s that contained emissions from small fires. The contributions of OBB to
two particulate pollution events in 2010 and 2012 were analyzed with brute-force
method. Attributed to varied OBB emissions and meteorology, the average
contribution of OBB to $PM_{10}$ concentrations in June 8-14 2012 was estimated at
37.6% (56.7 μg/m$^3$), larger than that in June 17-24, 2010 at 21.8% (24.0 μg/m$^3$).
Influences of diurnal curves of OBB emissions and meteorology on air pollution
caused by OBB were evaluated by designing simulation scenarios, and the results
suggested that air pollution caused by OBB would become heavier if the
meteorological conditions were unfavorable, and that more attention should be paid to
the OBB control at night. Quantified with Monte-Carlo simulation, the uncertainty of





traditional bottom-up inventory was smaller than that of FRP-based one. The
percentages of CRBF and emission factors were the main source of uncertainty for the
two approaches, respectively. Further improvement on CTM for OBB events would
help better constraining OBB emissions.
1.    Introduction

Open biomass burning (OBB) is an important source of atmospheric particulate
matter (PM) and trace gases including methane ($CH_4$), non-methane volatile organic
compounds (NMVOCs), carbon monoxide (CO), carbon dioxide ($CO_2$), oxides of
nitrogen ($NO_X$), sulfur dioxide ($SO_2$), and ammonia ($NH_3$) (Andreae and Merlet, 2001;
van der Werf et al., 2010; Wiedinmyer et al., 2011; Kaiser et al., 2012; Giglio et al.,
2013, Qiu et al., 2016; Zhou et al., 2017a). As it has significant impacts on air quality
and climate (Crutzen and Andreae, 1990; Cheng et al., 2014; Hodzic and Duvel, 2018),
it is important to understand the amount, temporal variation and spatial pattern of
OBB emissions.
Various methods have been used to estimate OBB emissions, including
traditional bottom-up method that relied on surveyed amount of biomass burning
(traditional bottom-up method), the method based on burned area or fire radiative
power (BA or FRP method), and emission constraining with chemistry transport
modeling (CTM) and observation (constraining method). In the traditional bottom-up
method that was most frequently used, emissions were calculated as a product of crop
production level, the ratio of straw to grain, percentage of dry matter burned in fields,
combustion efficiency, and emission factors (Streets et al., 2003; Cao et al., 2007;
Wang and Zhang, 2008; Zhao et al., 2012; Xia et al., 2016, Zhou et al., 2017a). The
BA or FRP method was developed along with progress of satellite observation
technology. BA was detected through remote sensing, and used in OBB emission
calculation combined with ground biomass density burned in fields, combustion
efficiency and emission factor. As the burned area of each agricultural fire was usually
small and difficult to be detected, this method could seriously underestimate the
emissions (van der Werf et al., 2010; Liu et al., 2015). In the FRP-based method, fire
radiative energy (FRE) was calculated with FRP at over pass time of satellite and the
diurnal cycle of FRP. The mass of crop residues burned in the field (CRBF) were then
obtained based on combustion conversion ratio and FRE, and emissions were





calculated as a product of the mass of CRBF and emission factor (Kaiser et al., 2012;
Liu et al., 2015). In the constraining method, the observed concentrations of
atmospheric compositions were used to constrain OBB emissions with CTM
(Hooghiemstra et al., 2012; Krol et al., 2013; Konovalov et al., 2014). The spatial and
temporal distributions of OBB emissions were derived from information of fire points
from satellite observation. Although varied methods and data sources might lead to
discrepancies in OBB emission estimation, those discrepancies and underlying
reasons have seldom been thoroughly analyzed in previous studies. Moreover, few
studies applied CTM to evaluate emissions obtained from different methods, thus the
uncertainty and reliability of OBB emission estimates remained unclear.
Due to growth of economy and farmers' income, a large number of crop straws
were discharged and burned in field, and OBB (which refers to crop straws burned in
fields in this paper) became an important source of air pollutants in China (Streets et
al., 2003; Shi and Yamaguchi 2014; Qiu et al., 2016; Zhou et al., 2017a). It brings
additional pressure to the country, which is suffering poor air quality (Richter et al.,
2005; van Donkelaar et al., 2010; Xing et al., 2015; Guo et al., 2017) and making
efforts to reduce pollution (Xia et al., 2016; Zheng et al., 2017). Located in the eastern
China, the Yangtze River Delta (YRD) region including the city of Shanghai and the
provinces of Anhui, Jiangsu and Zhejiang is one of China's most developed and
heavy-polluted regions (Ran et al., 2009; Xiao et al., 2011; Cheng et al., 2013, Guo et
al., 2017). Besides intensive industry and fossil fuel combustion, YRD is also an
important area of agriculture production, and frequent OBB events aggravated air
pollution in the region (Cheng et al., 2014).
In this study, therefore, we chose YRD to develop and evaluate high resolution
emission inventories of OBB with different methods. Firstly, we established OBB
emission inventories for 2005-2012 using the traditional bottom-up method (the
percentages of CRBF for 2013-2015 were currently unavailable), and inventories for
2005-2015 using FRP-based and constraining methods. The three inventories were
then compared with each other and other available studies with different spatial scales,
in order to discover the differences and their origins. Meanwhile, the three inventories
were evaluated using Models-3 Community Multi-scale Air Quality (CMAQ) system
and available ground observations. Contributions of OBB to ambient PM pollution
during two typical OBB events in 2010 and 2012 were evaluated through brute-force
method, and influences of meteorology and diurnal curves of OBB emissions on PM



pollution caused by OBB were also analyzed by designing simulation scenarios.
Finally, uncertainties of the three OBB inventories were analyzed and quantified with
Monte-Carlo simulation, and possible ways to improve OBB emission estimation in
were accordingly suggested.

**2. Data and methods**
**2.1 Traditional bottom-up method**
Annual OBB emissions in YRD were calculated by city from 2005 to 2012 using
the traditional bottom-up method with following equations:
$$E_{(i,y),j} = \sum_k \left( M_{(i,y),k} \times EF_{j,k} \right) \qquad (1)$$
$$M_{(i,y),k} = P_{(i,y),k} \times R_k \times F_{(i,y)} \times CE_k \qquad (2)$$
where $i$ and y indicate city and year (2005-2012), respectively; $j$ and $k$ represent
species and crop type, respectively; $E$ is the emissions, metric ton (t); $M$ is the mass of
CRBF, Gg; $EF$ is the emission factor, g/kg; $P$ is the crop production, Gg; $R$ is the ratio
of grain to straw (dry matter); $F$ is the percentage of CRBF; and $CE$ is the combustion
efficiency.
As summarized in Table S1 in the supplement, emission factors were obtained
based on a comprehensive literature review, and those developed in China were
selected preferentially. The mean value was used if various emission factors could be
obtained. When the emission factors for one crop straw were not obtained, the mean
value of the others was used instead. Annual production of crops at city level was
taken from statistical yearbooks (NBS, 2013). The ratios of straw to grain for different
crops were obtained from Bi (2010) and Zhang et al. (2008), and the combustion
efficiencies for different crop were obtained from Wang et al. (2013), as provided in
Table S2 in the supplement. Without officially reported data, the percentages of CRBF
were estimated to be half of the percentages of unused crop residues, following Su et
al. (2012). In Jiangsu, the percentages of unused crop residues were officially reported
for 2008, 2011 and 2012, while data for other years were unavailable. In this work,
therefore, the percentages of CRBF were assumed to be constant before 2008 and to
decrease by same rate (-15.2%) from 2008 to 2011, since a provincial plan was made
in 2009 to increase the utilization of straw (JPDRC and SMAC, 2009). Similarly, the
percentages of CRBF for Shanghai were assumed to be constant before 2008 and to





decrease by same rate (-16.8%) from 2008 to 2012. Without any official plans
released, in contrast, constant percentages of CRBF were assumed for Zhejiang and
Anhui before 2011, and that for 2012 was taken from NDRC (2014). We applied
uniform percentages of CRBF for cities within a province attributed to lack of detailed
information at city level, as summarized in Table S3 in the supplement. OBB
emissions after 2012 were not calculated with the traditional bottom-up method,
attributed to lack of information on percentages of CRBF and unused crop residues
for corresponding years.

**2.2 FRP-based method**

Similar to traditional bottom-up method, OBB emissions of FRP-based method
were calculated by multiplying the mass of CRBF and emission factors of various
pollutants, but mass of CRBF were derived from FRP instead of government-reported
data. As the burned crop types could not be identified with FRP, uniform emission
factors were applied for different crop types (Randerson et al., 2015; Liu et al., 2016;
Qiu et al., 2016), as provided in Table S4 in the supplement.
The mass of CRBF was calculated with the following equation:
$M = FRE \times CR$ (3)
where $M$ represents the mass of CRBF, kg; $CR$ represents the combustion conversion
ratio from energy to mass (kg/MJ); and $FRE$ represents the total released radiative
energy in an active fire pixel obtained from satellite observation (MJ). We used a
combustion ratio ($CR$) of 0.41 $\pm$ 0.04 (kg/MJ) based on the results of Wooster et al.
(2005) in the field and Freeborn et al. (2008) in the laboratory. Diurnal cycle of FRP
from crop burning was assumed to follow a Gaussian distribution. Following Vermote
et al. (2009) and Liu et al. (2015), $FRE$ was calculated using a modified Gaussian
function as below:
$FRE = \int FRP = \int_{0}^{24} FRP_{\text{peak}} \left( b + e^{-\frac{(t-h)^2}{2\sigma^2}} \right) dt$ (4)
$FRE_{\text{peak}} = \dfrac{FRP_t}{\left[ b + e^{-\frac{(t-h)^2}{2\sigma^2}} \right]}$ (5)
where $FRP_{peak}$ is the peak fire radiative power in the fire diurnal cycle; $t$ is the
overpass time of satellite; and $b$, $\sigma$, and $h$ represent the background level of the diurnal
cycle, the width of fire diurnal curve, and the peak hour (local time, LT), respectively.





FRP data were taken from MODIS Global Monthly Fire Location Product
(MCD14ML) which provides data from both the Terra and Aqua satellites (Davies et
al., 2009). The active fire data in MCD14ML were derived from Terra with overpass
times at approximately 10:30 AM and 10:30 PM LT and Aqua satellite with overpass
times at 1:30 AM and 1:30 PM LT. The fire products provided the geographic
coordinates of fire pixels (also known as fire points), overpass times, satellites and
their FRP values. The land cover dataset (GlobCover2009) was used to define
croplands (European Space Agency and Universit éCatholique de Louvain, 2011).
Parameters $b,\ \sigma$, and $h$ from 2005 to 2015 were calculated using the inter-annual
Terra to Aqua (T/A) FRP ratios provided in Table S5 in the supplement:
$b = 0.86r^2 - 0.52r + 0.08$           (6)
$\sigma = 3.89r + 1.03$           (7)
$h = -1.23r + 14.57 + \varepsilon$           (8)
where $r$ represents the average T/A FRP ratio. Following Liu et al. (2015), we added a
parameter $\varepsilon$ (4h) to modify $FRP_{peak}$ hour (h) of the diurnal curve, and the modified
FRP diurnal curves could better represent observed FRP temporal variability than the
original, as shown in Figure S1 in the supplement. As a result, FRE was calculated to
range from $1.49 \times 10^6$ MJ in 2009 to $1.95 \times 10^6$ MJ in 2005, with a mean value of
$1.74 \times 10^6$ MJ for YRD region (Table S5).

## 201    2.3 Constraining method

CTM and observation of ground particle matter (PM) concentrations were
applied in constraining OBB emissions given the potentially big contribution of OBB
to particle pollution for harvest seasons (Fu et al., 2013; Cheng et al., 2014; Li et al.,
2014). To characterize the non-linearity between emissions and concentrations, an
initial inventory including OBB and other anthropogenic sources was applied in CTM,
and the response of PM concentrations to emissions was calculated by changing OBB
emissions by a certain fraction (5% in this study) in the model. We defined a response
coefficient as the ratio of relative change in PM concentrations to that in OBB
emissions. Simulated PM concentrations were then compared with available
observation, and the mass of CRBF and OBB emissions of all species were corrected
combining the obtained response coefficient and the discrepancy between observed
and simulated PM concentrations. The corrected emissions were applied again in



CTM and the process (including recalculation of response coefficient) repeated until
the discrepancies between observation and simulation was small enough (the value of
I in equation (9) is less than 0.1% in this study). To limit the potential uncertainty in
emissions from other sources, the differences between simulated and observed PM
concentrations for non-OBB event period were included in the analysis:
$$ I = \left| \frac{\sum\limits_{x,i} S_{x,i} - \sum\limits_{x,i} Q_{x,i} \times N_i}{\sum\limits_{x,i} O_{x,i}} - 1 \right| \tag{9} $$

where $x$ and $i$ stand for the time (time interval of simulation is hour) and city,
respectively; $O$ is the observed PM concentration; $S$ and $Q$ are the simulated PM
concentration with and without OBB emissions, respectively; and $N$ is the normalized
mean bias (NMB) for non-OBB event period.

As primary particles emitted from OBB are almost fine ones, ambient $PM_{2.5}$

concentrations were commonly observed to account for large fractions of $PM_{10}$ during
the OBB event. Figure S2 shows the observed concentrations of $PM_{2.5}$ and $PM_{10}$ at
Caochangmen station in Nanjing (the capital of Jiangsu) in June 2012, and the
average mass ratio of $PM_{2.5}$ to $PM_{10}$ reached 79% during the OBB event in June 8-14,
2012. The ratios might be even higher in northern YRD where most fire points were
detected. As ground $PM_{2.5}$ concentrations were unavailable in most cities of northern
YRD before 2013, we expected that $PM_{10}$ was an appropriate indicator for OBB
pollution, and observed $PM_{10}$ concentrations were used to constrain OBB emissions
instead in this study. The daily mean $PM_{10}$ concentrations of all cities were derived
from the officially reported Air Pollution Index (API) by China National
Environmental Monitoring Center (http://www.cnemc.cn/). The conversion from API
scores to $PM_{10}$ concentrations is discussed in the Supplement.

Figure 1 illustrated the monthly variations of fire occurrences in 2010 and 2012

(panels a1 and a2, respectively), spatial patterns of fire points (panels b1 and b2) in
June 2010 and 2012, city-level $PM_{10}$ concentrations in YRD region in June 2010 and
2012 (panels c1 and c2), and temporal variations of daily fire occurrences in June
2010 and 2012 (panels d1 and d2). From 2005 to 2012, most OBB activities were
found in June 2010 and 2012 and northern YRD was the region with the intensive fire
counts. Accordingly $PM_{10}$ concentrations in northern YRD cities were higher than
those in more developed and industrialized cities in the eastern YRD (e.g., Shanghai,



Suzhou, Wuxi, and Changzhou), because emissions of OBB overwhelmed those from
other sources (Li et al., 2014; Huang et al., 2016). Therefore we constrained OBB
emissions with observed $PM_{10}$ concentrations in northern YRD cities including
Xuzhou, Lianyungang, Fuyang, Bengbu, Huainan, Hefei, Chuzhou and Bozhou.
Suggested by the monthly and daily distribution of fire counts (Figure 1a and 1d), two
strong OBB events were defined for June 17-24, 2010 and June 8-14, 2012, and other
days in June of 2010 and 2012 were defined as non-OBB event period. For other
years, the OBB emissions were first scaled from the constrained emissions in 2010
and 2012 with the ratios of FRE for the given year to that for 2010 and 2012
respectively, and then calculated as average of the two.
Traditional bottom-up method was used to calculate the initial emission input for
all species (NMVOCs emission factor was taken from FRP-based method instead as
the value in bottom-up method (Li et al., 2007) did not contain oxygenated VOCs). In
contrast to application of uniform percentage of CRBF within one province, however,
percentage of CRBF for each city was calculated based on that in whole YRD and the
fraction of FRP in the city to total YRD FRP, to make the spatial distribution of OBB
emissions consistent with that of FRP all over YRD region:
$$F_{(i,y)} = \frac{FRP_{(i,y)}}{FRP_{(YRD,y)}} \times \frac{\sum\limits_{k} P_{(YRD,y),k}}{\sum\limits_{k} P_{(i,y),k}} \times F_{(YRD,y)} \tag{10}$$

where $i$ and $k$ represent city and crop type, respectively; $y$ indicates the year (2010 and
2012); $F$, $P$, and $FRP$ are the percentage of CRBF, crop production, and fire radiative
power, respectively. The initial percentage of CRBF for total YRD ($F_{(YRD,y)}$ in eq (10))
was expected to have limited impact on the result and it was set at 10%, smaller than
those in previous studies (Streets et al., 2003; Cao et al., 2007; Wang and Zhang, 2008;
Zhao et al., 2012; Xia et al., 2016, Zhou et al., 2017a).
**2.4 Temporal and spatial distributions**
The spatial and temporal patterns of OBB emissions in the three inventories were
determined according to the FRP of agricultural fire points. The emissions of $m$-th
grid in region $u$ on $n$-th day in year $y$ were calculated using equation (11):
$$E_{(m,n),j} = \frac{FRP_{(m,n)}}{FRP_{(u,y)}} \times E_{(u,y),j} \tag{11}$$

where $FRP_{(m,n)}$ is the FRP of $m$-th grid on $n$-th day; $FRP_{(u,y)}$ and $E_{(u,y),j}$ are the total
FRP and OBB emissions of species $j$ for region $u$ in year y, respectively. The region $u$



indicates city for FRP-based and constraining method, while it indicates province for
traditional bottom-up method since uniform percentages of CRBF was applied within
the same province in the method.

**2.5 Configuration of air quality modeling**

The Models-3 Community Multi-scale Air Quality (CMAQ) version 4.7.1 was
applied to constrain OBB emissions and to evaluate OBB inventories with different
methods. As shown in Figure 2, one-way nested domain modeling was conducted, and
the spatial resolutions of the two domains were set at 27 and 9 km respectively in
Lambert Conformal Conic projection, centered at ($110^o$E, $34^o$N) with two true
latitudes 25 and 40° N. The mother domain (D1, 180×130 cells) covered most parts
of China, Japan, North and South Korea, while the second domain (D2, 118×97 cells)
covered the whole YRD region. OBB inventories developed in this work were applied
in D2. Emissions from other anthropogenic sources in D1 and D2 were obtained from
the    downscaled    Multi resolution    Emission    Inventory    for    China    (MEIC,
http://www.meicmodel.org/) with an original spatial resolution of 0.25°×0.25°.
Population density was applied to relocate MEIC to each modeling domain. Biogenic
emission inventory was from the Model Emissions of Gases and Aerosols from
Nature developed under the Monitoring Atmospheric Composition and Climate
project (MEGAN MACC, Sindelarova et al., 2014), and the emission inventories of
Cl, HCl and lightning $NO_X$ were from the Global Emissions Initiative (GEIA, Price et
al., 1997). Meteorological fields were provided by the Weather Research and
Forecasting Model (WRF) version 3.4, and the carbon bond gas-phase mechanism
(CB05) and AERO5 aerosol module were adopted. Other details on model
configuration and parameters were given in Zhou et al. (2017b).
Meteorological parameters of WRF model were compared with the observation
dataset of US National Climate Data Center (NCDC), as summarized in Table S6 in
the Supplement. For June 2012, the average biases between the two datasets were
0.01 m/s for wind speed, 7 degree for wind direction, 0.91 K for temperature and
3.1% for relative humidity. The analogue numbers were 0.06 m/s, 9.84 degree, 0.64 K
and 2.99% respectively for June 2010. Simulated daily $PM_{10}$ concentrations were
compared with observation for non-OBB event period in June 2010 and 2012 in Table
S7 in the supplement. The average of normalized mean biases (NMB) and normalized
mean errors (NME) were -19.9% and 38.9% for 17 YRD cities in June 2010, and


-22.8% and 33.9% for 22 cities in June 2012, respectively. As shown in Figure S3 in
the supplement, moreover, simulated hourly $PM_{10}$ and $PM_{2.5}$ concentrations were in
good agreement with observations at four air quality monitoring sites in YRD during
non-OBB event period in June 2012. The comparison thus implied the reliability of
emission inventory of anthropogenic origin used in this work, while underestimation
might occur indicated by the negative NMB.

### 3. Results and discussions
**3.1 OBB emissions estimated with the three methods**

OBB emissions estimated with the traditional bottom-up method for 2005-2012

were shown in Table S8 in the supplement. As emission factors were assumed
unchanged during the period, similar inter-annual trends were found for all species
and $CO_2$ was selected as a representative species for further discussion. As shown in
Figure 3, $CO_2$ emissions from traditional bottom-up method were estimated to
decrease from 23000 in 2005 to 19973 Gg in 2012, with a peak value of 27061 Gg in
2008. In contrast, the number of fire points in YRD farmland increased from 7158 in
2005 to 17074 in 2012. The fire counts detected from satellite thus did not support the
effectiveness of OBB restriction by government in YRD before 2013. Table S9 in the
supplement presents the annual OBB emissions derived from FRP-based method for
2005-2015 in YRD region. Associated with fire counts, $CO_2$ emissions were estimated
to grow by 119.7% from 2005 to 2012, with the largest and the second largest annual
emissions calculated at 19977 and 12718 Gg for 2012 and 2010, respectively (Figure
3). Similar temporal variability was found for fire counts, which increased by 138.5%
from 2005 to 2012, with the most and the second most counts found at 17074 and
12322 for 2012 and 2010, respectively.

With the constraining method, as shown in Figure S4 in the supplement, the ratio

of constrained mass of CRBF for 2012 to 2010 was 1.51, clearly lower than the ratios
of original FRE (1.75) but close to the ratio of modified FRE for 2012 to 2010 (1.57).
The comparison suggested that modified FRE better reflect the OBB activity in YRD
than original FRE. In order to make the ratio of FRE for the two years be closer to the
ratio of constrained mass of CRBF, an improved method was developed for
calculating the FRE. Given the possible variation of $FRP_{peak}$ hour between years, we
obtained the diurnal cycle of total FRP of YRD based on Gaussian fitting as shown in





342 Figure S5 in the supplement. The ratio of FRE for 2012 to 2010 was recalculated at

343 1.54, further closer to the ratio of constrained mass of CRBF. Therefore the ratios of

344 FRE for another given year to 2012 and 2010 were calculated with this improved

345 method, and were then applied to emission scaling for that year. The constrained OBB

346 emissions from 2005 to 2015 were summarized in Table 1. As shown in Figure 3, the

347 inter-annual trend in constrained emissions was similar with those in fire counts and

348 FRP-based emissions but different from that in emissions with the traditional

349 bottom-up method. It is usually difficult to collect accurate percentages of CRBF from

350 bottom-up method, as it demands intensive investigation in the rural areas. In addition,

351 the percentages of CRBF were not updated for each year, and same percentages were

352 commonly applied for years without sufficient data support from local surveys.

353  The constrained $CO_2$ emissions for Jiangsu, Anhui, Zhejiang and Shanghai were

354 calculated at 5790, 4699, 1104 and 419 Gg in 2005, accounting for 48.2%, 39.1%,

355 9.2% and 3.5% of total OBB emissions in YRD, respectively. The analogue numbers

356 for 2012 were 7345, 16159, 2574 and 394 Gg, and 27.7%, 61.0%, 9.7% and 1.5%,

357 respectively. Jiangsu and Anhui were found to contribute largest to OBB emissions in

358 YRD for 2005 and 2012, respectively. In the traditional bottom-up method, however,

359 Anhui was estimated to contribute largest for both years. City-level OBB emissions

360 estimated with the three methods were summarized in Tables S10-S12 in the

361 supplement. With the constraining method, in particular, the largest $CO_2$ emissions

362 were found in Suzhou (1708 Gg) of Anhui, Lianyungang (1578 Gg) and Xuzhou

363 (1401 Gg) of Jiangsu in 2005, accounting for 14.2%, 13.1% and 11.7% of the total

364 emissions, respectively. In 2012, Suzhou, Bozhou of Anhui, and Xuzhou of Jiangsu

365 were identified as the cities with the largest emissions, with the values estimated at

366 5007, 2433, and 2109 Gg, respectively. Depending on distribution of fire points, the

367 shares of OBB emissions by city were close between the constraining and FRP-based

368 method, and large emissions concentrated in the north of YRD. Based on the surveyed

369 percentages of CRBF and crop production, in contrast, the emission shares by city in

370 the traditional bottom-up method were clearly different from the other two, and

371 emissions concentrated in Anhui cities with high crop production level.

372  The average annual emissions of $CO_2$ for 2005-2011 with traditional bottom-up

373 method were 87.0% larger than those in constraining method and the emissions for

374 2012 was 24.6% times smaller than those in constraining method. Given the same

375 sources of emission factors for all species except NMVOCs, the discrepancies of OBB



emissions for most species between constraining and traditional bottom-up methods come from the activity levels (i.e., percentages of CRBF and crop production). The average annual constrained emissions from 2005 to 2015 were larger than those derived by FRP-based method for all species except EC, $CH_4$ and $NH_3$, since the average annual mass of CRBF from the constraining method were 36.9% larger than those from the FRP-based method for those years, as shown in Figure S6 in the supplement.

The percentage of CRBF is an important parameter to judge OBB activity and to estimate emissions. Besides the investigated values applied in traditional bottom-up approach, the percentages of CRBF were recalculated based on the constrained emissions at provincial level and were shown in Figure S7 in the supplement. The largest and smallest percentages of CRBF in the whole YRD region were estimated at 18.3% in 2012 and 8.1% in 2006, respectively. The inter-annual trend in percentages of CRBF for YRD was closest to that for Anhui province, as the province dominated the crop burning in the region. The different inter-annual trends by province were strongly influenced by agricultural practice and government management. Agricultural practice could be associated with income level and mechanization level. Increased income would lead to more crop residues discarded and burned in the field, while development of mechanization would lead to less. The constrained percentages of CRBF for Shanghai increased from 2005 to 2007 and declined after 2007, while those for Jiangsu decreased from 2005 to 2008 and increased after 2008. Increasing trends were found for the percentages of CRBF for Anhui and Zhejiang from 2005 to 2012, and they might result largely from growth of farmers' income. The percentages of CRBF for all provinces except Zhejiang decreased significantly in 2008, attributed largely to the measures of air quality improvement for Beijing Olympic Games. Shanghai was the only one with its percentage of CRBF significantly reduced in 2010, resulting mainly from the air pollution control for Shanghai World Expo in that year. Compare to those in the bottom-up method, the constrained percentages of CRBF for Anhui and Jiangsu for all the years except 2012 were smaller, leading to smaller constrained OBB emissions than the bottom-up ones in those years.

The constrained percentages of CRBF and straw yields for 2012 were shown by city in Figure S8 in the supplement, and clear inconsistency in spatial distributions can be found. The percentage of CRBF was not necessarily high for a city with large straw production. For instance, the straw production of Yancheng was larger than



most other cities, but its percentage of CRBF was 5.7% and lower than most other
cities. Through linear regression, correlation coefficient was calculated at only 0.06
between constrained percentage of CRBF and straw yield at city level. The poor
correlation between them suggested that large uncertainty could be derived if uniform
percentage of CRBF was applied to calculate OBB emissions for cities within given
province, as what we did in the traditional bottom-up methodology.

**3.2  Evaluation of the three OBB inventories with CMAQ**

Figures 4 and 5 illustrate the observed daily averaged and simulated hourly $PM_{10}$
concentrations for selected YRD cities in June 17-25, 2010 and June 8-14, 2012,
respectively. Four cases, i.e., emission inventory without and with OBB emissions
estimated using the three methods, were included. The simulated $PM_{10}$ concentrations
without OBB emissions were significantly lower than observation for all cities,
implying that OBB was an important source of airborne particulates during the two
periods. Simulations with OBB emissions derived from the three methods performed
better than those without OBB emissions for most cities during June17-25, 2010 and
all cities during June 8-14, 2012. The best performance was found for simulations
with constrained OBB emissions in most cities during the two periods, and the high
$PM_{10}$ concentrations were generally caught by CTM for the concerned OBB events. In
2010, for example, the observed high concentrations were caught by CTM with the
constrained emissions in Lianyungang on June 21-23, and Fuyang and Huainan on
June 19-21. In 2012, the high concentrations were caught in Xuzhou on June 12-14,
Lianyungang on June 13-14, Fuyang on June 11-12, Bozhou on June 10 and Chuzhou
on June 11-12. The results indicated that fire points could principally capture the
temporal and spatial distribution of OBB emissions. Overestimation still existed in
CTM with the constrained OBB emissions for the cities with intensive fire points (e.g.,
Xuzhou, Bozhou and Fuyang in 2012 and Bengbu in 2010), while underestimation
commonly existed for cities with fewer fire points (e.g., Hefei, Chuzhou and Huainan
in 2010 and 2012). Due to limitation of MODIS observation, fires at moderate to
small scales could not be fully detected (Giglio et al., 2003; Schroeder et al., 2008),
thus the spatial allocation of OBB emissions based on FRP could possibly result in
more emissions than actual in areas with intensive fire points.
The NMB and NME between observed and simulated $PM_{10}$ concentrations are
shown in Table 2. Among all the cases, the NMB and NME with the constrained OBB



emissions were smaller than most of those with other OBB emissions, implying the
best guess of OBB emissions obtained through the constraining method combining
CTM and ground observation. The simulated $PM_{10}$ concentrations using FRP-based
OBB emissions were smaller than observation for the two periods, due mainly to the
underestimated mass of CRBF. The results indicated that the OBB emissions might be
underestimated in FRP-based method in 2010 and 2012, since many small fires in
YRD were undetected in MODIS active fire detection products. The probability of
MODIS detection was strongly dependent upon the temperature and area of the fire
being observed. The average probability of detection for tropical savanna was 33.6%
when the temperature of fire was between 600 and 800℃ and the area of fire was
between 100 and 1000 $m^2$ (Giglio et al., 2003). In YRD region, on one hand, the fire
temperature of crop residue burned in fields was relatively low. On the other hand,
nearly 100 farmers were possibly located in a single $1 \times 1$ km MODIS pixel (Liu et al.,
2015), and a famer commonly owned croplands of several hundred square meters.
Therefore many fire pixels in YRD might not be detected, leading to underestimation
in the total FRE. The simulated $PM_{10}$ concentrations with the traditional bottom-up
OBB emissions were higher than observation in 2010 but lower in 2012. The results
thus implied the growth in OBB emissions from 2010 to 2012 could not be captured
by traditional bottom-up method, attributed partly to application of unreliable
percentage of CRBF.

**3.3 Comparisons of different studies and methods**

To explore the influence of data and methods on OBB emission estimation, we

selected CO to compare emissions in this work and other national and global
inventories for YRD, given the similar emission factors of CO applied in various
studies. CO emissions from the three methods in this work were compared with
GFASv1.0 (Kaiser et al., 2012), GFEDv3.0 (van der Werf et al., 2010), GFEDv4.1
(Randerson et al. 2015), Wang and Zhang (2008), Huang et al. (2012), Xia et al. (2016)
and Zhou et al. (2017a), as shown in Figure 6. The emissions from Wang and Zhang
(2008), Huang et al. (2012), Xia et al. (2016) and Zhou et al. (2017a) were derived by
traditional bottom-up method, while GFASv1.0, GFEDv3.0 and GFEDv4.1 were
based on FRP and BA methods. In particular, the emissions from small fires were
included in GFEDv4.1. Similar inter-annual variations were found for emissions



derived based on FRP measurement including the constrained and FRP-based emissions in this work, GFAS v1.0, and GFED v4.1, while those of GFEDv3.0 and Xia et al. (2016) were different. The percentages of CRBF were assumed unchanged during the studying period in Xia et al. (2016), thus the temporal variation of OBB emissions were associated with the change in annual straw production.

The constrained CO emissions in this work were smaller than other studies using the traditional bottom-up method (Wang and Zhang, 2008; Huang et al., 2012; Xia et al., 2016) and larger than those based on burned area and FRP derived from satellite (GFEDv3.0; GFASv1.0; GFEDv4.1). In particular, the average annual constrained emissions from 2005 to 2012 were 3.9, 0.5 and 15.0 times larger than those in GFASv1.0, GFEDv4.1s and GFEDv3.0, respectively. The constrained emissions were closest to GFED v4.1s that included small fires. As described in Section 3.2, the area of farmland belonging to individual farmers was usually small, and small fires were expected to be important sources of OBB emissions in YRD. GFEDv4.1s might still underestimate OBB emissions due to the omission errors for the small fires in MODIS active fire detection products (Schroeder et al., 2008). In addition, the constrained CO emission for 2013 was 31.5% larger than those by Qiu et al. (2016) calculated based on burned area from satellite observations. The average annual CO emissions from 2005 to 2012 by the constraining method were 57.2% smaller than Xia et al. (2016), and the constrained emissions for 2006 were respectively 27.6% and 56.9% smaller than those by Huang et al. (2012) and Wang and Zhang (2008). It implied again that the traditional bottom-up method might overestimate OBB emissions during the period. Moreover, the discrepancy in estimations between Huang et al. (2012) and Wang and Zhang (2008) for the same year resulted mainly from application of different percentages of CRBF, implying that calculation of OBB emissions was sensitive to the parameter with the bottom-up approach.

The spatial distribution of the constrained emissions in this work and those in GFASv1.0, GFEDv3.0 and GFEDv4.1s were illustrated in Figure 7. Intensive OBB emissions in GFEDv3.0 were mainly found in parts of Anhui, Jiangsu and Shanghai, while the constrained emissions, GFEDv4.1s and GFASv1.0 emissions occurred in the most YRD regions in accordance with the distribution of fire points. Therefore, GFEDv3.0 might miss a large number of burned areas, leading to underestimation in emissions and bias in spatial distribution.





In order to understand the discrepancies between this work and other inventories
for different species, the emissions of 2010 derived from the three methods in this
study, GFASv1.0, GFEDv3.0, GFEDv4.1s and Xia et al. (2016) were summarized in
Table 3. Similar to CO, the constrained emissions in this work were larger than
GFASv1.0, GFEDv3.0 and GFEDv4.1s for most species except $NH_3$, but were smaller
than the estimates in Xia et al. (2016) and this study based on the bottom-up method
for all the species. In addition, the constrained emissions for most species were
smaller than the bottom-up estimates by Huang et al. (2012), Wang and Zhang (2008)
and Xia et al. (2016) for 2006. The comparison implied again that the FRP/BA-based
method might underestimate the OBB emissions attributed to the detection limit on
small fires. In contrast, application of the assumed/surveyed fraction of burned
biomass in the bottom-up method could often overestimate OBB emissions.
Resulting from the different sources of emission factors, the discrepancies
between studies or methods varied greatly by species. For $PM_{10}$ and $PM_{2.5,}$ as an
example, the emissions by Xia et al. (2016) were respectively 35.8% and 50.3%
higher than the constrained emissions in 2010. The discrepancies for $SO_2$ and $NO_X$
were larger: the emissions by Xia et al. (2016) were 4.7 and 3.1 times larger than our
constrained emissions, respectively. Moreover, the constrained NMVOCs emission
was 152.5 and 10.7 times larger than that of GFEDv3.0 and GFEDv4.1s in 2010,
respectively, partly because the emission factors of GFEDv3.0 and GFEDv4.1s did
not contain oxygenated VOCs. In contrast, the constrained $NH_3$ emissions were 4.7%
and 47.9% smaller than that of GFEDv3.0 and GFEDv4.1s, respectively. The
comparisons indicated that emission factors were important sources of uncertainties in
estimation of OBB emissions with different methods.

**3.4 Contribution of OBB to particulate pollution and its influencing factors**

The brute-force method (BFM, Dunker et al., 1996) was used to analyze the
contributions of OBB to particulate pollution for the two OBB events, June 17-24,
2010 and June 8-14, 2012. Simulated $PM_{10}$ concentrations with and without
constrained OBB emissions were compared, and the difference indicated the
contribution from OBB as shown by city in Figure 8. The average contribution in June
8-14, 2012 was estimated at 37.6% (56.7 μg/m$^3$) for 22 cities in YRD, and the
contribution for June 17-24, 2010 was smaller at 21.8 % (24.0 μg/m$^3$) for 17 cities.
Our result for 2012 was nearly the same as that for 5 YRD cities in 2011 (37.0%) by





Cheng et al. (2014). Using the BFM method, the contribution of OBB emissions to
$PM_{10}$ concentrations were estimated to increase by 136.3% from 2010 to 2012 in this
work, and the enhancement was larger than that of OBB emissions (50.8%). Therefore,
factors other than emissions (e.g., meteorology) could also play an important role in
elevating the contribution of OBB to ambient particle pollution. For example, the
average precipitation in June 8-14, 2012 was 36% smaller than that in June 17-24,
2010, exaggerating the particle pollution during OBB event.

The average contributions of OBB for 2012 were estimated at 55.0% (98.4

$\mu g/m^3$), 36.4% (58.0 $\mu g/m^3$), 23.6% (12.9 $\mu g/m^3$), and 14.4% (11.2 $\mu g/m^3$) for 6 cities
of Anhui, 10 cities of Jiangsu, 5 cities of Zhejiang and Shanghai, respectively. For
individual cities, large contributions of OBB for 2012 were found in Xuzhou, Bozhou,
Fuyang, and Lianyungang located in the north YRD, reaching 82.3% (284.3 $\mu g/m^3$),
75.2% (207.5 $\mu g/m^3$), 71.9% (134.7 $\mu g/m^3$) and 63.5% (96.2 $\mu g/m^3$), respectively.
Similarly, large contributions for 2010 were found in Lianyungang, Fuyang and
Bozhou reaching 63.3% (69.8 $\mu g/m^3$), 58.2% (71.9 $\mu g/m^3$) and 78.8% (53.6 $\mu g/m^3$),
respectively. In general the spatial distribution of OBB contributions to $PM_{10}$ mass
concentrations was similar with that of fire points, confirming the rationality of
constraining OBB emissions with observed $PM_{10}$ concentration in cities in north
Anhui and Jiangsu.

To explore the influence of meteorology on air pollution caused by OBB, we

simulated $PM_{10}$ concentrations for June 8-14 (PE1) and June 22-28 2012 (PE2) with
varied meteorology conditions but fixed OBB emissions (i.e., constrained emissions
for June 8-14, 2012). Poorer meteorology conditions during PE1 were found than PE2.
The average wind speed in PE1 was 2.4 m/s, 17% lower than that in PE2. The average
wind direction in PE1 was 168.3°, close to south with polluted air in land. In contrast,
the average wind direction in PE2 was 118.3°, close to east with clean air from the
ocean. The average precipitation in PE2 was 6.8 mm, 28% larger than that in PE1. As
shown in Figure 9, the average contribution of OBB to $PM_{10}$ concentrations for 22
cities in YRD region was estimated at 56.7 $\mu g/m^3$ for PE1, 23% larger than that for
PE2, and the contributions in most cities were much larger for PE1 than those for PE2,
except for Bozhou and Fuyang. The comparisons suggested that air pollution caused
by OBB would exaggerate under poorer meteorology conditions. To reduce air
pollution caused by OBB in harvest season in YRD, therefore, more attention should





be paid to the OBB restriction on those days with unfavorable meteorology conditions
such as calm wind and rainless period.
To further analyze the influence of diurnal variation of emissions on air pollution
caused by OBB, we simulated $PM_{10}$ concentrations on June 17-24 2010 with various
diurnal curves of OBB emissions (i.e., those for 2010 and 2012). Constrained
emissions were applied in the simulation. As shown in Figure 10, the contributions of
OBB to $PM_{10}$ concentrations based on diurnal curve of 2012 were larger than those
based on 2010 for almost all YRD cities, and the average contribution for the 17 cities
was calculated at 28.6 $\mu g/m^3$ based on diurnal curve of 2012, 10% larger than that
based on 2010. The contribution in Bozhou changed most (1.37 times larger with
2012 curve), while those in Shanghai, Huzhou and Shaoxing changed least. The time
of peak value for OBB emissions in 2012 was 2.5 hours later than 2010, indicating
that the fraction of OBB emissions at night for 2012 would be larger than that for
2010. As the diffusion condition for air pollutants at night was usually worse than that
during daytime, more OBB emissions at night would elevate its contribution to
particle pollution. In the actual fact, the supervision of OBB prohibition was usually
conducted by government during daytime, thus some farmers burned more crop
residues at night to avoid the punishment. To improve the air quality in harvest season
in YRD, more attention should be paid to the OBB restriction at night.
**3.5 Uncertainty analysis**
The uncertainties of OBB emissions estimated with bottom-up and FRP-based
methods were quantified by species using a Monte-Carlo simulation for 2012. A total
of 20,000 simulations were performed and the uncertainties were expressed as 95%
confidence intervals (CIs) around the central estimates. The parameters contributing
most to OBB emission uncertainty were also identified according to their contribution
to the variance in Monte-Carlo simulation.
For traditional bottom-up method, parameters included crop productions,
percentages of CRBF, straw to grain ratios, combustion efficiencies, and emission
factors. Crop production was directly taken from official statistical yearbooks (NBS,
2013) and its uncertainty was expected to be limited and thereby not included in the
analysis. As the percentage of CRBF was determined at half of the percentage of
unused crop residues, its uncertainty was set at -100% to +100%. The combustion
efficiencies were assumed within an uncertainty range of 10% around the mean value





according to de Zarate (2005) and Zhang et al. (2008). The uncertainties of emission factors were obtained from original literatures where they were derived. If the emission factor was derived from a single measurement, normal distribution was applied with the standard deviation directly taken from that measurement. If the emission factor was derived from multiple measurements and the samples were insufficient for data fitting, uniform distribution was tentatively applied with a conservative strategy to avoid possible underestimation of uncertainty: The uncertain range of the emission factor would be expanded according to Li et al. (2007) if the range obtained originally from the multiple measurements was smaller than that in Li et al. (2007). Summarized in Table S13 in the supplement was a database for emission factors and percentages of CRBF, with their uncertainties indicated by probability distribution function (PDF). As shown in Table 4, the uncertainties of OBB emissions for 2012 based on the traditional bottom-up method were estimated at -56% to +70%, -56% to +70%, -50% to +54%, -54% to +73%, -49% to +58%, -48% to +59%, -46% to +73%, -48% to +60%, -47% to +87%, -59% to +138% and -51% to +67% for $PM_{10}$, $PM_{2.5}$, EC, OC, $CH_4$, NMVOCs, CO, $CO_2$, $NO_X$, $SO_2$ and $NH_3$, respectively. For most species, the percentages of CRBF contributed largest to the uncertainties of OBB emissions, while emission factors were more significant to $SO_2$ uncertainty.

For the FRP-based method, parameters included total FRE, combustion conversion ratio and emission factors. The uncertainty of total FRE was associated with FRP value, MODIS detection resolution, and the methodology used to calculate FRE per fire pixel. Indicated by Freeborn et al. (2014), the coefficient of variation of MODIS FRP was 50% for a fire pixel, but it declined to smaller than 5% for the aggregation of over 50 MODIS active fire pixels. Give the large number of fire pixels for in YRD (more than 17000 in 2012), FRP was expected to contribute little to uncertainty of total FRE and could thus be ignored. Due to limitation of MODIS resolution, small fires could not be fully detected and the number of fire pixel could be underestimated by 300% on crop-dominant areas (Schroeder et al., 2008), therefore the uncertainty of number of fire pixel was assumed to be 0 to +300%. The method used to calculate FRE based on single fire pixel assumed that fire lasted one day. Given the small cropland owned by one farmer in YRD, individual fire normally lasted several hours, and FRE could be overestimated. As the total FRE in FRP-based method was estimated 2.6 times larger than that from constraining method based on the same number of the fire pixel, we tentatively assumed the uncertainty range of



FRE for one fire pixel at -72% to 0%. The uncertainty of total FRE was then
estimated at -17% to +154% (95% CIs) based on the principle that total FRE was
calculated as the number of fire pixel multiplied by average FRE. The uncertainty of
combustion conversion ratio was derived from Wooster et al. (2005) and Freeborn et
al. (2008), while those of emission factors taken from Akagi et al. (2011). As a result,
the uncertainties of FRP-based inventory were estimated at -77% to +274%, -63% to
+244%, -78% to +281%, -78% to 276%, -83% to +315%, -63% to +243%, -52% to
+223%, -21% to +164%, -82% to +303%, -78% to +279%, and -82% to +302% for
$PM_{10}$, $PM_{2.5}$, EC, OC, $CH_4$, NMVOCs, CO, $CO_2$, $NO_X$, $SO_2$ and $NH_3$ in 2012,
respectively. Emission factors contributed most to the uncertainties of emissions for
all species except $CO_2$.
The uncertainty of constrained emissions could hardly be calculated by
Monte-Carlo simulation, as the results were associated with CTM performance. In
general, CTM performance could be influenced by emission estimates for
anthropogenic sources other than OBB, chemistry mechanism of CTM and temporal
and spatial distribution of OBB emissions. Emission inventory of anthropogenic
sources that incorporates the best available information of individual plants was
expected to improve the CTM performance at the regional or local scale (Zhou et al.,
2017b). The influence of chemistry mechanism came mainly from secondary organic
carbon (SOC) modeling. According to the Cheng et al. (2014) and Chen et al. (2017),
the mass fraction of SOC to $PM_{10}$ could reach 10% during the OBB event in YRD,
and that part might not be well constrained with the approach we applied in this work.
Similar to FRP-based method, moreover, temporal and spatial distribution of OBB
emissions based on FRP might not be entirely consistent with the reality, due to the
omission errors in the MODIS active fire detection products and the limited times of
satellite overpass as discussed earlier.
The uncertainties of OBB emissions with traditional bottom-up method were
estimated smaller than those with FRP-based method, and uncertainties for $CO_2$ and
CO were usually smaller than other species in both methods attributed mainly to
fewer variations in their emission factors. OBB emission estimation with traditional
bottom-up method could be improved if more accurate percentages of CRBF are
obtained, and that with FRP-based method could be improved when the omission
error of satellite and the uncertainties of emission factors are reduced. Efforts should



also be recommended on improvement of CTM for better constraining the OBB
emissions.

## 4. Conclusions

Taking YRD in China as an example, we developed OBB emission inventories
with traditional bottom-up, FRP-based and constraining methods, and analyzed the
discrepancies between them and the underlying reasons. The simulated $PM_{10}$
concentrations through CMAQ with constrained emissions were closest to available
observation, implying the improvement of emission estimation with this method. The
inter-annual variations in emissions with FRP-based and constraining methods were
similar with the fire counts, while that with traditional bottom-up method was not.
The contrast indicated that the bottom-up method could not capture the actual
inter-annual trend of OBB emissions. The emissions of all species except NMVOCs
based on bottom-up method might be overestimated in most years, attributed mainly
to application of elevated percentages of CRBF. The emissions with FRP-based
method might be underestimated in 2005-2015, attributed to the omission errors in the
MODIS active fire detection products and thereby to the underestimation in mass of
CRBF. Compared with other inventories at different spatial scales, similar temporal
variations of CO emissions were found for the constrained emissions, FRP-based
emissions in this work, and emissions in GFASv1.0 and GFEDv4.1s. The constrained
CO emissions in this work were usually smaller than those in the bottom-up
inventories derived both in this work and other studies, but larger than those in
FRP-based inventories derived both in this work and other studies. The comparison
again demonstrated that the bottom-up method might overestimate OBB emissions in
YRD and the FRP-based method might underestimate them. The OBB contributions
to particulate pollution in typical episodes were analyzed using the Brute-force
method in CMAQ modeling. The OBB emissions in 2012 were 51% larger than those
in 2010, while its contribution to average $PM_{10}$ mass concentrations was estimated to
increase by 136% from 2010 to 2012. The enhanced contribution of OBB was not
attributed only to the growth in OBB emissions but was also partly caused by the
meteorology. Quantified with a Monte-Carlo framework, the uncertainties of OBB
emissions with traditional bottom-up method were smaller than those with FRP-based
method. The uncertainties of traditional bottom-up and FRP-based emission





estimations were mainly from the percentages of CRBF and emission factors, respectively. Further improvement on CTM for OBB events would help better constraining OBB emissions.

Limitations remained in this study. Given the difficulty in field investigation, the annual CRBF used in the traditional bottom-up method was obtained from limited studies and it could not correctly reflect the actual OBB activity. The reliability of FRP-based estimation depended largely on the detection resolution of the satellite. In YRD where the burned areas of individual fires were small, many fires could not be detected by MODIS. The accuracy of constrained emissions depended largely on CTM performance and the spatial and temporal distributions of OBB emissions derived from satellite-observed FRP. Therefore FRP-based and constraining method may be improved if more reliable fire information is obtained. In addition, more measurements on local emission factors for OBB are suggested in the future to reduce the uncertainty of emissions.

## Acknowledgements

This work was sponsored the National Key Research and Development Program of China (2016YFC0201507 and 2017YFC0210106), Natural Science Foundation of China (91644220 and 41575142), Natural Science Foundation of Jiangsu (BK20140020), and Special Research Program of Environmental Protection for Common wealth (201509004). The MCD14ML data were provided by LANCE FIRMS operated by the NASA/GSFC/Earth Science Data and Information System (ESDIS) with funding provided by NASA/HQ.

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



**FIGURE CAPTIONS**

**Figure 1. (a) Monthly variations of fire occurrences in 2010 and 2012, (b) spatial patterns of fire points in June 2010 and June 2012, (c) PM$_{10}$ concentrations for city-level in YRD in June 2010 and June 2012, and (d) temporal variations of daily fire occurrences in June 2010 and 2012. City abbreviations FY, BZ, BB, HN, HF, CZ(a), XZ, LYG, NJ, YZ, ZJ, TZ, NT, CZ, WX, SZ, HZ(a), JX, HZ, SX, NB, SH indicate is Fuyang, Bozhou, Bengbu, Huainan, Hefei, Chuzhou, Xuzhou, Lianyungang, Nanjing, Yangzhou, Zhenjiang, Taizhou, Nantong, Changzhou, Wuxi, Suzhou, Huzhou, Jiaxing, Hangzhou, Shaoxing, Ningbo, and Shanghai).**

**Figure 2. Model domain and locations of 43 meteorological monitoring sites.**

**Figure 3. Fire counts and CO$_2$ emissions estimated with traditional bottom-up, FRP-based and constraining methods for YRD 2005-2012.**

**Figure 4. Observed 24-hour averaged PM$_{10}$ concentrations and simulated hourly PM$_{10}$ concentrations without OBB emissions (No_OBB) and with OBB emissions based on traditional bottom-up (Traditional_OBB), FRP-based (FRP_OBB) and constraining (Constrained_OBB) methods in Lianyungang, Fuyang, Bozhou, Bengbu, Huainan, Hefei, and Chuzhou during June 17-25, 2010.**

**Figure 5. Observed 24-hour averaged PM$_{10}$ concentrations and simulated hourly PM$_{10}$ concentrations without OBB emissions (No_OBB) and with OBB emissions based on traditional bottom-up (Traditional_OBB), FRP-based (FRP_OBB) and constraining (Constrained_OBB) methods in Xuzhou, Lianyungang, Fuyang, Bozhou, Bengbu, Huainan, Hefei, and Chuzhou during June 8-14, 2012.**

**Figure 6. Annual CO emissions from OBB in YRD obtained in this work and other studies from 2005 to 2012.**

**Figure 7. Spatial distributions of CO emissions from OBB obtained in this work (constraining method), GFAS v1.0, GFED v3.0 and GFED v4.1s in 2010.**

**Figure 8. The contribution of OBB to PM$_{10}$ concentrations for different YRD cities during OBB events in June 2010 and 2012.**

**Figure 9. PM$_{10}$ concentrations contributed by OBB for different YRD cities in Jun 8-14 (PE1) and June 22-28 (PE2), 2012.**

**Figure 10. PM$_{10}$ concentrations contributed by OBB for different YRD cities**





**based on the diurnal variations of 2010 and 2012 in Jun 8-14, 2010.**



**TABLES**
**Table 1. Constrained OBB emissions from 2005 to 2015 in YRD (Unit: Gg).**

|  | $PM_{10}$ | $PM_{2.5}$ | EC | OC | $CH_4$ | NMVOCs | CO | $CO_2$ | NOx | $SO_2$ | $NH_3$ |
|---|---|---|---|---|---|---|---|---|---|---|---|
| 2005 | 175.7 | 153.7 | 4.4 | 38.7 | 32.1 | 420.3 | 670.2 | 12011.2 | 22.2 | 2.7 | 4.1 |
| 2006 | 171.3 | 149.9 | 4.3 | 37.8 | 31.3 | 409.9 | 653.7 | 11716.7 | 21.7 | 2.6 | 4.0 |
| 2007 | 219.1 | 191.7 | 5.5 | 48.3 | 40.0 | 524.2 | 835.9 | 14981.9 | 27.7 | 3.4 | 5.1 |
| 2008 | 176.7 | 154.6 | 4.4 | 39.0 | 32.3 | 422.8 | 674.3 | 12085.2 | 22.3 | 2.7 | 4.1 |
| 2009 | 178.8 | 156.4 | 4.5 | 39.4 | 32.6 | 427.7 | 682.0 | 12223.3 | 22.6 | 2.8 | 4.2 |
| 2010 | 257.9 | 225.7 | 6.5 | 58.3 | 47.6 | 624.5 | 987.7 | 17720.3 | 33.0 | 4.0 | 6.1 |
| 2011 | 188.9 | 165.3 | 4.7 | 41.7 | 34.5 | 452.0 | 720.7 | 12917.7 | 23.9 | 2.9 | 4.4 |
| 2012 | 389.0 | 340.4 | 9.6 | 83.6 | 70.2 | 919.4 | 1478.6 | 26473.6 | 48.6 | 6.0 | 9.0 |
| 2013 | 260.7 | 228.1 | 6.5 | 57.5 | 47.6 | 623.8 | 994.7 | 17828.1 | 33.0 | 4.0 | 6.1 |
| 2014 | 332.4 | 290.8 | 8.3 | 73.3 | 60.7 | 795.2 | 1268.1 | 22729.0 | 42.0 | 5.1 | 7.8 |
| 2015 | 109.9 | 96.1 | 2.8 | 24.2 | 20.1 | 262.9 | 419.3 | 7514.6 | 13.9 | 1.7 | 2.6 |




**Table 2. Model performance statistics for concentrations of PM$_{10}$ from**
**observation and CMAQ simulation without OBB emissions (No_OBB) and with**
**OBB emissions based on traditional bottom-up (Traditional_OBB), FRP-based**
**(FRP_OBB) and constraining methods (Constrained_OBB) for the two OBB**
**events of June 2010 and 2012.**

|  | June 2010 | | June 2012 | |
|---|---|---|---|---|
|  | NMB | NME | NMB | NME |
| No_OBB | -47% | 50% | -60% | 68% |
| Traditional_OBB | 11% | 44% | -16% | 45% |
| FRP_OBB | -33% | 41% | -45% | 52% |
| Constrained_OBB | -16% | 37% | -10% | 45% |

Note: NMB and NME were calculated using following equations ($P$ and $O$ indicate the results
from modeling prediction and observation, respectively):
$$NMB = \frac{\sum_{i=1}^{n}(P_i - O_i)}{\sum_{i=1}^{n}(O_i)} \times 100\% \; ; \;\; NME = \frac{\sum_{i=1}^{n}|P_i - O_i|}{\sum_{i=1}^{n}(O_i)} \times 100\% \quad .$$



**Table 3. OBB emissions in YRD derived from this work and other studies in**
**2010 (Unit: Gg).**

|  | PM$_{10}$ | PM$_{2.5}$ | EC | OC | CH$_4$ | NMVOCs | CO | CO$_2$ | NOx | SO$_2$ | NH$_3$ |
|---|---|---|---|---|---|---|---|---|---|---|---|
| Traditional (this work) | 362.4 | 317.1 | 9.3 | 85.7 | 67.9 | 154.9 | 1391.8 | 24978.0 | 47.0 | 5.4 | 8.7 |
| FRP-based (this work) | 57.8 | 50.6 | 6.4 | 18.5 | 46.5 | 412.5 | 820.1 | 12718.0 | 24.9 | 3.2 | 17.7 |
| Constrained (this work) | 257.9 | 225.7 | 6.5 | 58.3 | 47.6 | 624.5 | 987.7 | 17720.3 | 33.0 | 4.0 | 6.1 |
| GFASv1.0 | - | 17.8 | 1.0 | 9.5 | 15.6 | 88.7 | 196.3 | 3097.8 | 5.1 | 1.0 | 3.1 |
| GFEDv3.0 | - | 3.5 | 0.2 | 1.7 | 3.2 | 4.1 | 39.4 | 701.6 | 1.1 | 0.2 | 6.4 |
| GFEDv4.1s | - | 33.6 | 4.0 | 12.4 | 31.3 | 53.2 | 548.3 | 8519.7 | 16.7 | 2.2 | 11.7 |
| Xia et al, (2016) | 350.2 | 339.3 | 14.8 | 137.8 | - | - | 1989.9 | 49835.1 | 134.3 | 22.6 | - |






**Table 4. The uncertainties of OBB emissions in YRD indicated as 95% CIs and**
**the top two parameters contributing most to emission uncertainties based on**
**traditional bottom-up and FRP-based methods for 2012. The percentages in the**
**parentheses indicate the contributions of the parameters to the variances of**
**emissions.**

| | Traditional bottom-up method | | FRP-based method | |
|---|---|---|---|---|
| $PM_{10}$ | -56%, +70% | PCRBF$^1_{Anhui}$(42%)<br>EF$_{wheat}$ (41%) | -77%, +274% | EF(76%)<br>AF$^2$ (11%) |
| $PM_{2.5}$ | -56%, +70% | PCRBF$_{Anhui}$(43%)<br>EF$_{wheat}$ (41%) | -63%, +244% | EF(65%)<br>NFP$^3$ (16%) |
| EC | -50%, +54% | PCRBF$_{Anhui}$(69%)<br>PCRBF$_{Jiangsu}$(11%) | -78%, +281% | EF(75%)<br>NFP (11%) |
| OC | -54%, +73% | PCRBF$_{Anhui}$(42%)<br>EF$_{rice}$ (37%) | -78%, +276% | EF(75%)<br>NFP (11%) |
| $CH_4$ | -49%, +58% | PCRBF$_{Anhui}$(65%)<br>PCRBF$_{Jiangsu}$(11%) | -83%, +315% | EF(79%)<br>NFP (9%) |
| NMVOCs | -48%, +59% | PCRBF$_{Anhui}$(64%)<br>PCRBF$_{Jiangsu}$(10%) | -63%, +243% | EF(65%)<br>NFP (16%) |
| CO | -46%, +73% | PCRBF$_{Anhui}$(62%)<br>PCRBF$_{Jiangsu}$(10%) | -52%, +223% | EF(57%)<br>NFP (19%) |
| $CO_2$ | -48%, +60% | PCRBF$_{Anhui}$(69%)<br>PCRBF$_{Jiangsu}$(10%) | -21%, +164% | NFP (44%)<br>AF (42%) |
| $NO_X$ | -47%, +87% | PCRBF$_{Anhui}$(51%)<br>EF$_{wheat}$ (23%) | -82%, +303% | EF(78%)<br>NFP (10%) |
| $SO_2$ | -59%, +138% | EF$_{wheat}$ (35%)<br>PCRBF$_{Anhui}$(27%) | -78%, +279% | EF(74%)<br>NFP (12%) |
| $NH_3$ | -51%, +67% | PCRBF$_{Anhui}$(55%)<br>EF$_{wheat}$ (12%) | -82%, +302% | EF(79%)<br>NFP (10%) |

[1] PCRBF, the percentage of crop residues burned in the field (the subscript indicates province); [2]
AF, the average FRE of fire pixels; [3] NFP, the number of fire pixels; [4] MCRBF, the mass of crop
residues burned in the field.



**Figure 1.**

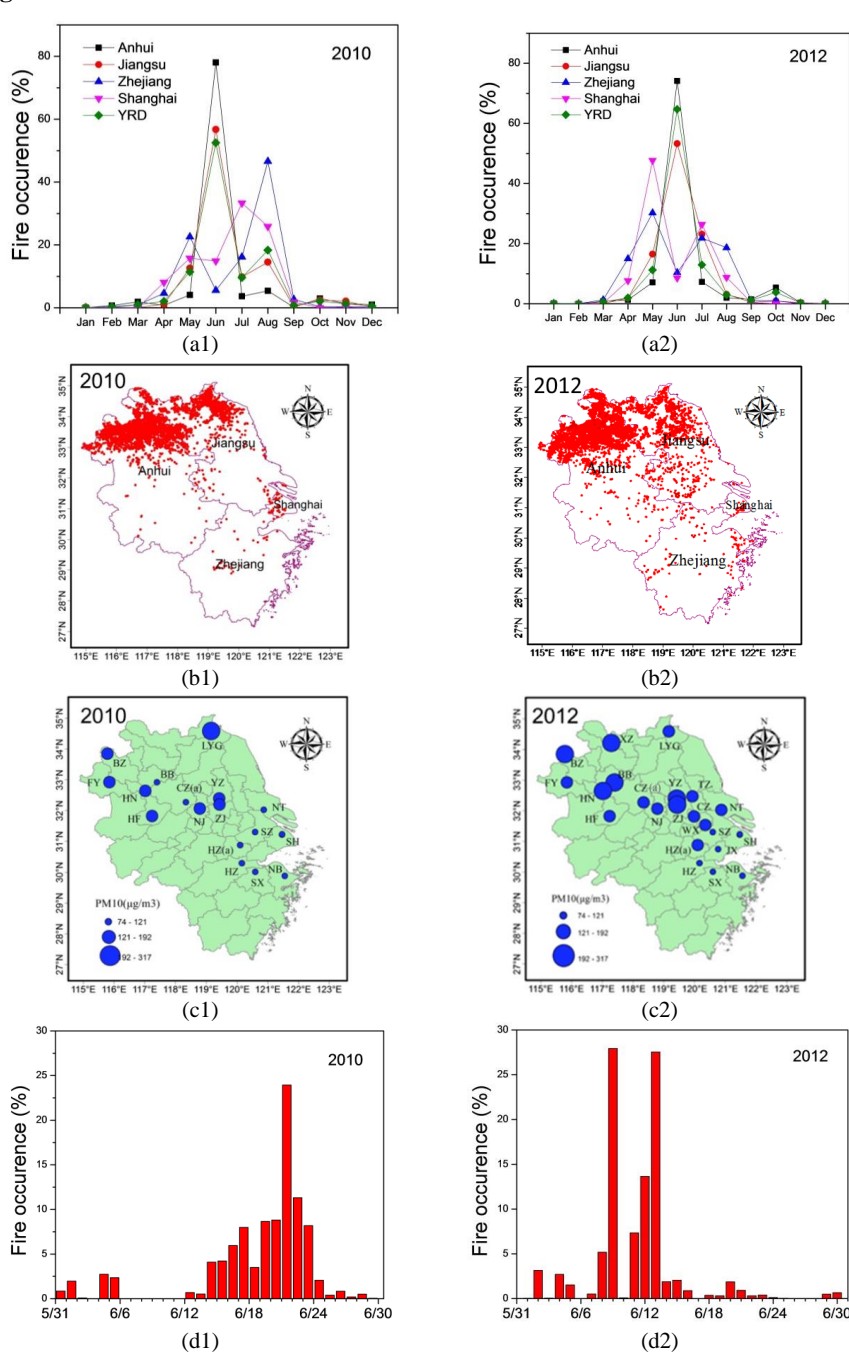



**Figure 2.**

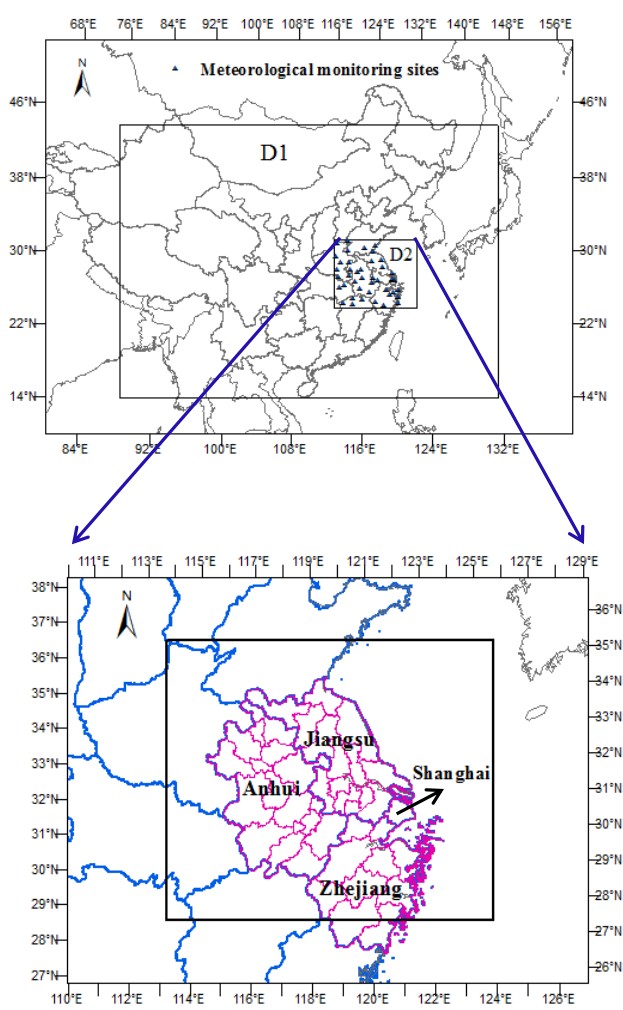






**Figure 3.**

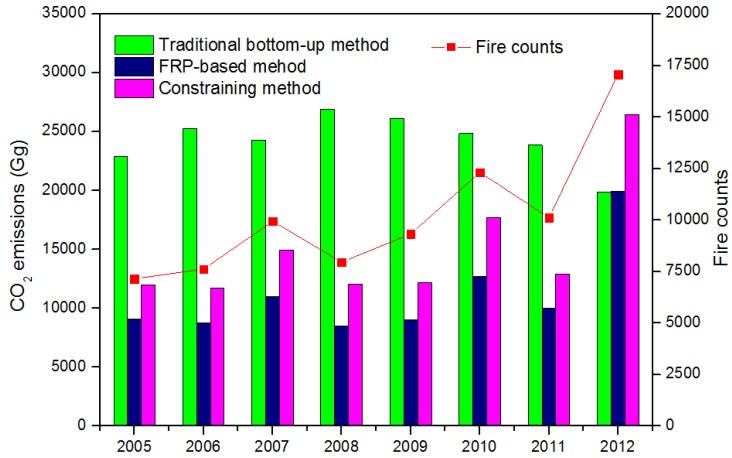







**Figure 4.**

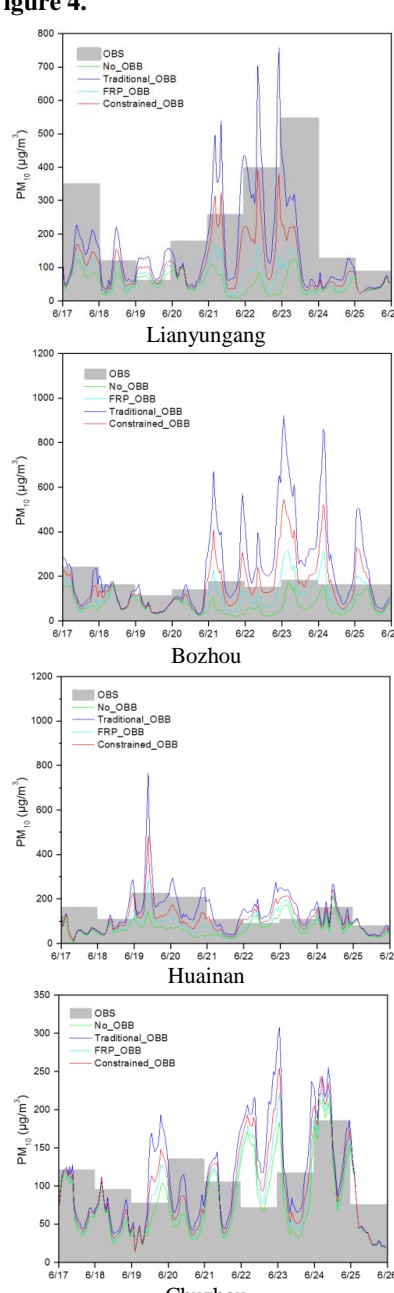

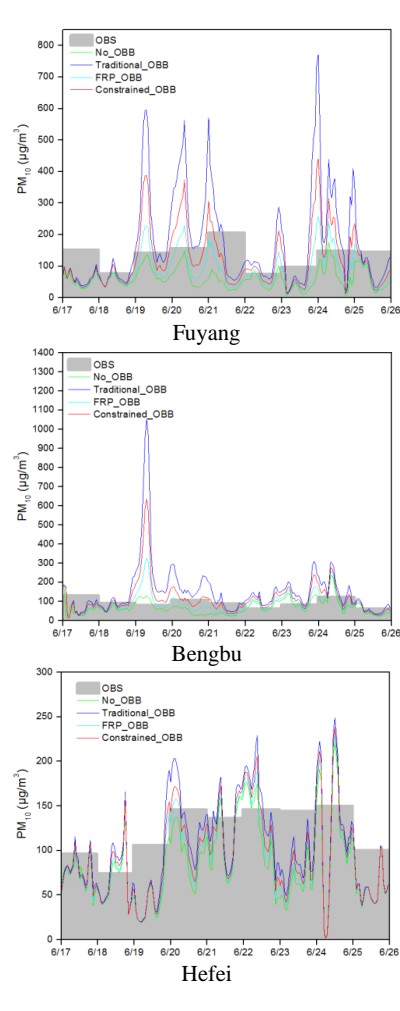







**Figure 5.**

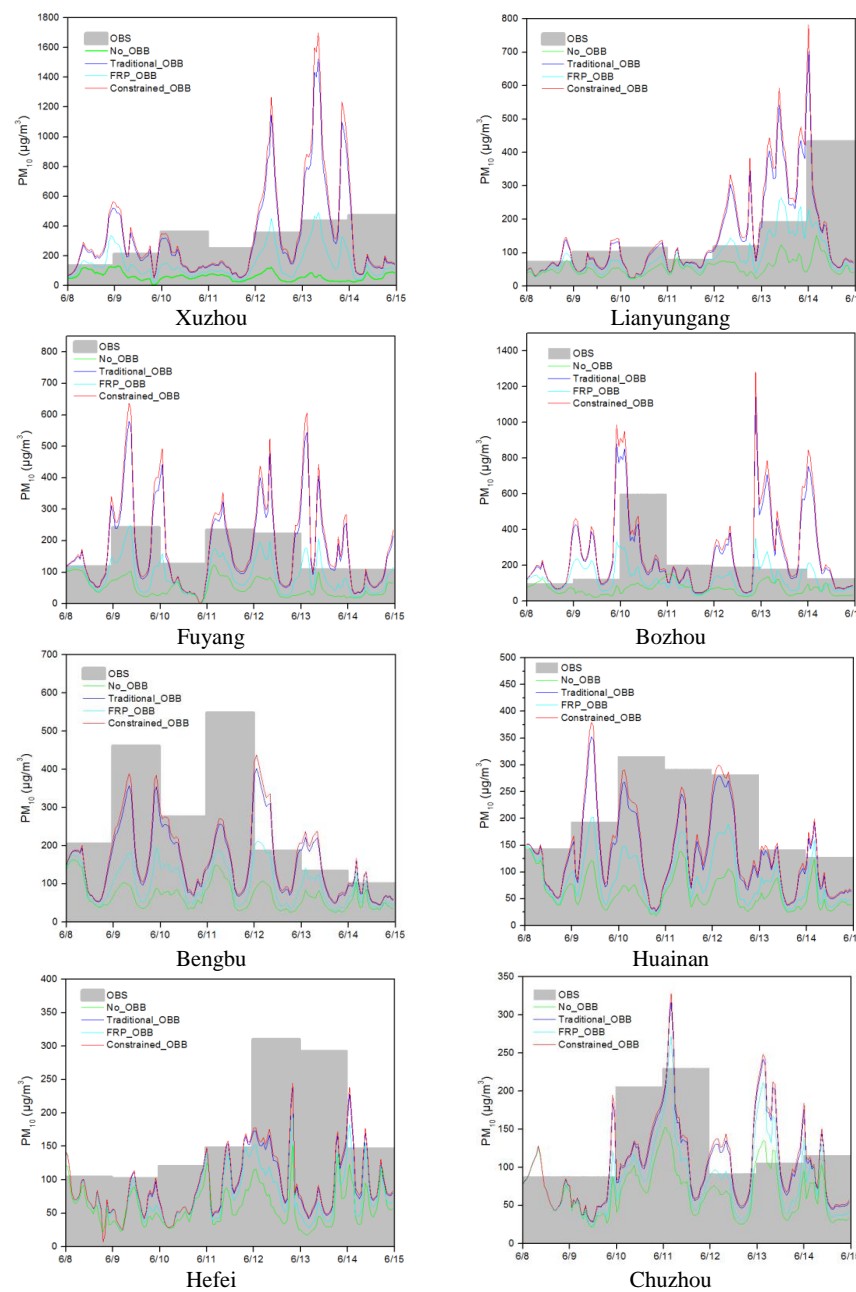





**Figure 6.**

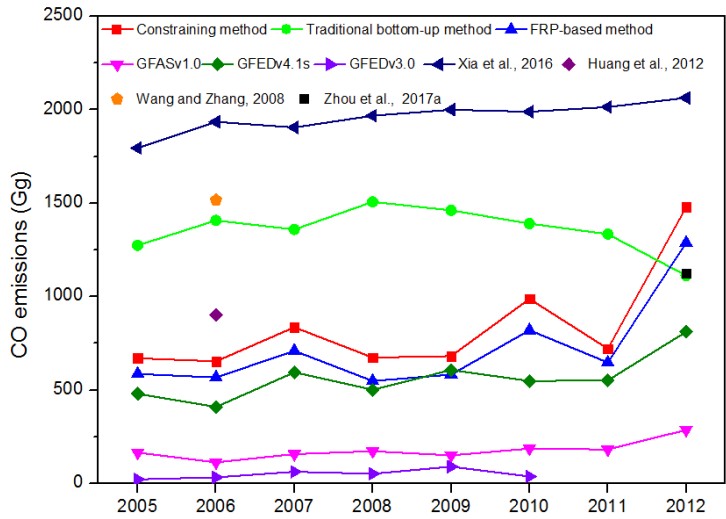





**Figure 7.**

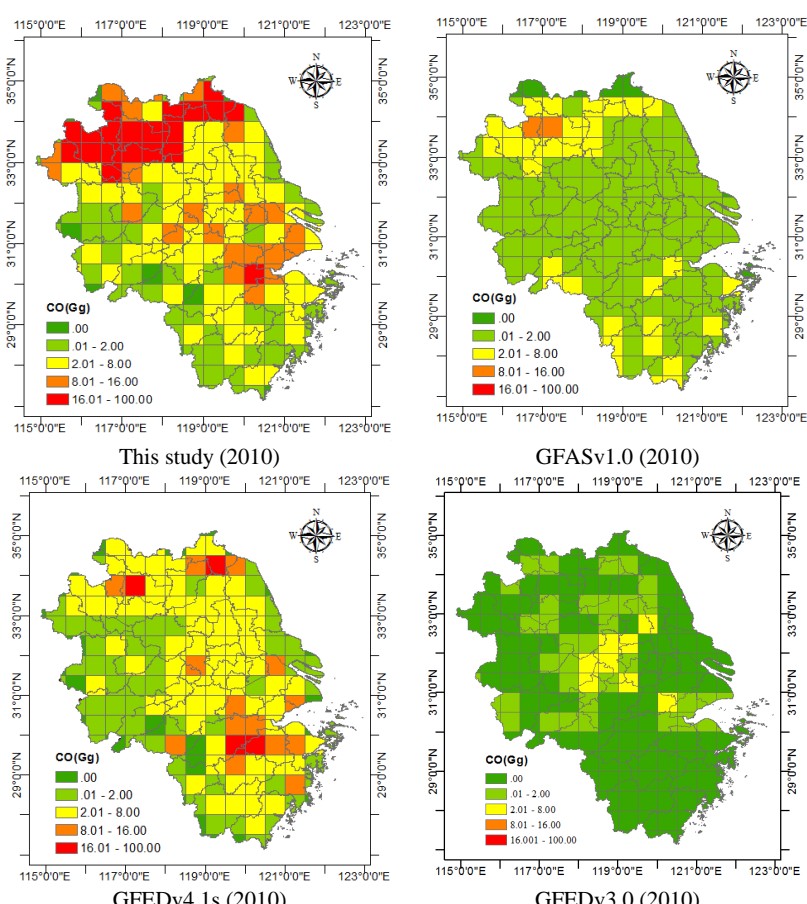





**Figure 8.**

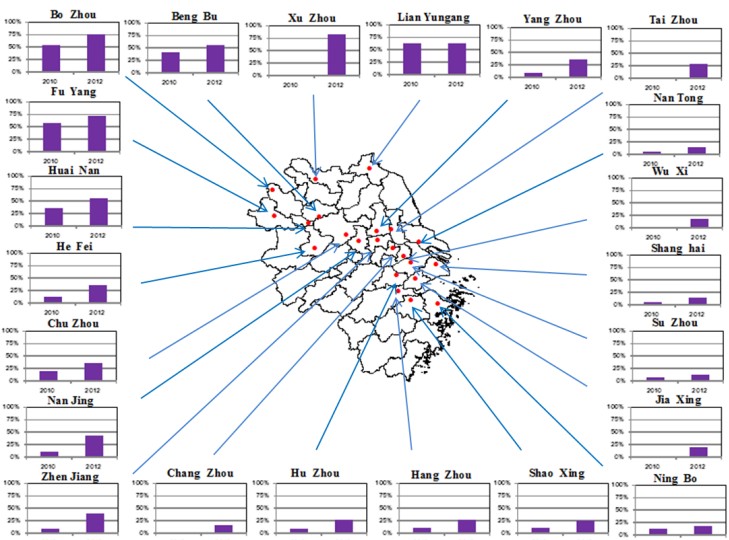





**Figure 9.**

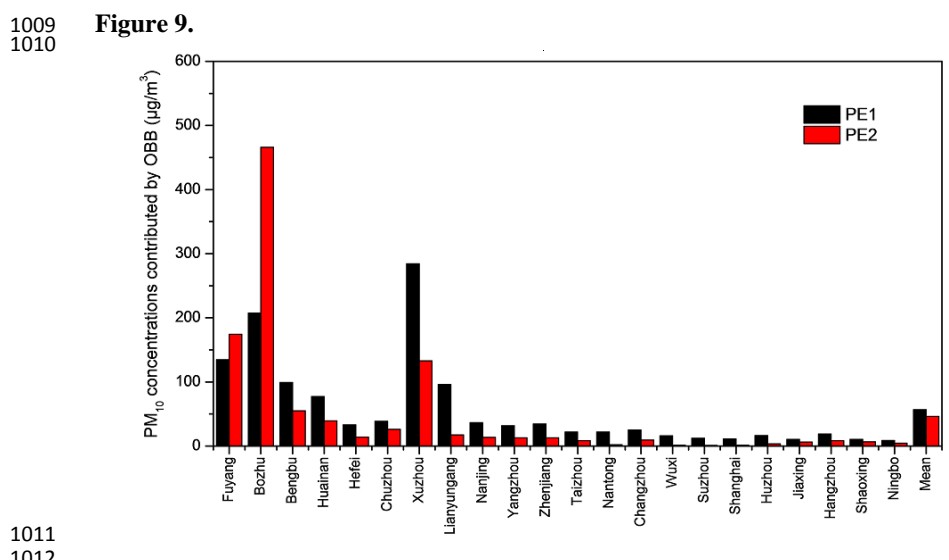






**Figure 10.**

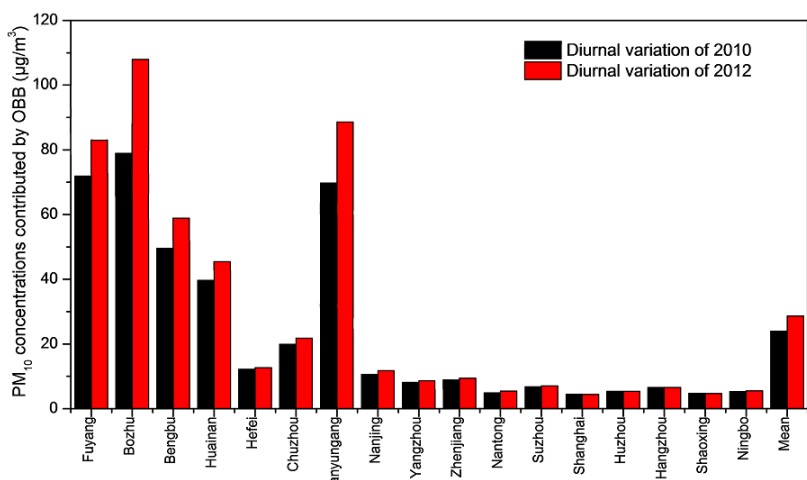
