# Peer review of "Quantification and evaluation of atmospheric pollutant emissions from open biomass burning with multiple methods: A case study for Yangtze River Delta region, China"

_Atmospheric Chemistry and Physics, 2018_

## Referee Comment (RC1) · Anonymous Referee #1 · 12 Sep 2018

This manuscript presents a very comprehensive study of historical trend of OBB emissions in YRD. I am very impressed by the large amounts of work done in this study. The presentation is also of high quality, and the structure is well organized. The constraining method is a little bit weak, but makes the story complete. I would suggest the authors improve the constraining method in future studies. The authors have acknowledged the weakness, which is great. I only have very minor comment for improvements. For constraining method, the correction is based on the comparisons of PM10, and the correction factor was applied to all other species. The authors should acknowledge

this limitation in the method section.

---

## Referee Comment (RC2) · Anonymous Referee #2 · 19 Sep 2018

This manuscript estimates the air pollutant emissions from open biomass burning (OBB) in Yangtze River Delta for 2005-2015 using traditional bottom-up, fire radiative power (FRP)-based, and constraining approaches, and analyzed the differences between those methods and their underlying reasons. The manuscript is generally well written. However, there are still some issues in the manuscript which authors shall pay attention to. So the paper cannot be accepted for publication before authors address the following comments.

1. As shown in Table S1 and Table S4, the authors use different emission factors for

[Figure]

OBB in bottom-up method and FRP-based method. I suggest same emission factors shall be used for both methods. This is why that for most air pollutants, emissions estimated by bottom-up method is higher than that by FRP-based but the emissions of NMVOC and NH3 from bottom-up method is much lower than that by FRP-based method.

2. The spatial resolutions of the two domains were set at 27 and 9 km respectively. 9km is kind of coarse resolution. How does this spatial resolution affect the CMAQ modeling results? Will you get a better model performance if you use a 3km resolution?

3. Considering that the PM emissions from OBB are mainly PM2.5, and the ambient PM10 is more affected by the local road dust emissions, it is not appropriate to only use PM10 concentration to evaluate the model performance and analyze the contribution of OBB. I think authors shall use both PM10, PM2.5, CO, NO2, SO2, OC, EC to do the model evaluation. At least PM2.5 shall be included considering that most Chinese cities release PM2.5 hourly concentrations since 2013. Although authors give a couple of figures in SI, this is not enough. Specifically, the correction based on the comparisons of PM10 cannot be used for all other species.

4. The model performance statistics for meteorological parameters shown in Table S6 and that for PM10 concentrations as shown in Table 2 shall include the benchmark of the evaluation.

5. For OBB, temporal allocation is very important. It is good to see the monthly variations of fire occurrence in Figure 1. However, the authors only give information for year 2010 and 2012, I wonder if the authors can provide such information for other years.

6. Figure 2 shall give the name of each city in the YRD. Otherwise it is difficult for readers to understand when author talk about Lianyungang, Fuyang, Shanghai, Suzhou, Wuxi, Changzhou, etc.

7. The color in Figure 4 is very difficult to read.

---

## Author Response (AR1)

**Main revisions and response to reviewers' comments**

Manuscript No.: acp-2018-701

Title: Quantification and evaluation of atmospheric pollutant emissions from open biomass burning with multiple methods: A case study for Yangtze River Delta region, China

Authors: Yang Yang, Yu Zhao

We thank very much for the valuable comments and suggestions from the two reviewers, which help us improve our manuscript. The comments were carefully considered and revisions have been made in response to suggestions. Following is our point-by-point responses to the comments and corresponding revisions.

**Reviewer #1**

*1. General comments: This manuscript presents a very comprehensive study of historical trend of OBB emissions in YRD. I am very impressed by the large amounts of work done in this study. The presentation is also of high quality, and the structure is well organized. The constraining method is a little bit weak, but makes the story complete. I would suggest the authors improve the constraining method in future studies. The authors have acknowledged the weakness, which is great. I only have very minor comment for improvements. For constraining method, the correction is based on the comparisons of PM10, and the correction factor was applied to all other species. The authors should acknowledge this limitation in the method section.*

**Response and revisions:**

We appreciate the reviewer's positive remarks on our manuscript. We thank the reviewer's suggestion and will improve the constraining method in future studies from following aspects. The method can be improved incorporating the observed ambient concentrations of multiple pollutants (e.g., $PM_{10}$, $PM_{2.5}$, OC and EC) if those concentrations with sufficient temporal and spatial resolution get available. Improvement on the results of constraining method can be expected if more reliable emission factors of biomass burning and improved and the emissions of other sources are obtained and applied in the study. For constraining method, the correction of activity level was based on the comparison between simulated and observed $PM_{10}$ concentrations, and the emissions of other species were then revised according to the changed activity level. In this method, the emission estimation of other species depends largely on the reliability of emission factors for $PM_{10}$ and those species. Large uncertainty may exist due to lack of sufficient domestic measurements. We take the reviewer's suggestion and acknowledge this limitation in the method section. **Corresponding revision was shown in lines 258-264 of Page 9 in the revised manuscript.**

**Reviewer #2**

*1. This manuscript estimates the air pollutant emissions from open biomass burning(OBB) in Yangtze River Delta for 2005-2015 using traditional bottom-up, fire radiative power (FRP)-based, and constraining approaches, and analyzed the differences between those methods and their underlying reasons. The manuscript is generally well written. However, there are still some issues in the manuscript which authors shall pay attention to. So the paper cannot be accepted for publication before authors address the following comments.*

**Response and revisions:**

We appreciate the reviewer's crucial and important comments. In general, the presentation of the work has been improved, based on specific comments/suggestion from the reviewer. Same emission factors as bottom-up method were applied to estimate the OBB emissions for 2010 based on FRP-based method, and the results were compared with those based on bottom-up method. Both $PM_{2.5}$ and $PM_{10}$ concentrations were used to evaluate the model performance and to analyze the contribution of OBB in June 7-13, 2014. The benchmarks of the evaluation for model performance and meteorological parameters were added in Table 2 and Table S6 in the supplement. We also take the reviewer's suggestion and provide the monthly variations of fire occurrence for other years in Figure S3 in the supplement. Details follow.

*2. As shown in Table S1 and Table S4, the authors use different emission factors for OBB in bottom-up method and FRP-based method. I suggest same emission factors shall be used for both methods. This is why that for most air pollutants, emissions estimated by bottom-up method is higher than that by FRP-based but the emissions of NMVOC and NH3 from bottom-up method is much lower than that by FRP-based method.*

**Response and revisions:**

We thank the reviewer's comment. In the bottom-up method, the masses of crop residues burned in the field (CRBF) for different crop species could be obtained, therefore the emission factors for different crop types were usually used. However, the masses of CRBF for different crop species in FRP-based method could not be obtained, and the emission factors based on burned area or fire radiative power (BA or FRP method) by other researchers (van der Werf et al., 2010, Kaiser et al., 2012; Liu et al., 2015; Randerson et al., 2018) were applied, ignoring the difference between crop types. In order to know the differences between the OBB emissions based on FRP-based and bottom-up methods with same emission factors, we followed the reviewer's comment and made an extra case: the emission factors applied in the bottom-up method were weighted with the masses of various crop types and used to estimate the OBB emissions for 2010 with the FRP-based method. The estimated OBB emissions (FRP-based (WSE)) were compared with the emissions based on bottom-up method as shown in Table 3. The OBB emissions for all species in FRP-based (WSE) were smaller than those derived by bottom-up method. The differences in OBB emissions between bottom-up and FRP-based (WSE) method were larger than 50% of those between the bottom-up and the original FRP-based method with different emission factors for most species. It indicated that the discrepancy in activity level contributed the most to the difference in OBB emissions between bottom-up and FRP-based method. **Corresponding revision was shown in lines 200-205 of Page 7 and lines 553-559 of Page 18 in the revised manuscript.**

*3. The spatial resolutions of the two domains were set at 27 and 9 km respectively. 9km is kind of coarse resolution. How does this spatial resolution affect the CMAQ modeling results? Will you get a better model performance if you use a 3km resolution?*

**Response and revisions:**

We thank the reviewer's comment. The model performance largely depends on the reliability of emission inventories. The emissions of other sources in this study were obtained from the downscaled the Multi resolution Emission Inventory for China (MEIC) with an original spatial resolution of $0.25°×0.25°$. The model performance with a finer resolution might not necessarily be better since the emissions were probably not distributed in the correct grids in finer resolution with a simple spatial interpolation (Zheng et al., 2017). Improvement on emission inventory with the underlying data carefully compiled and analyzed is important to achieve better model performance with high-resolution chemistry transport modeling. Our previous study by Zhou et al. (2017) evaluated the downscaled MEIC and improved local emission inventory with CMAQ modeling at a 3 km resolution in southern Jiangsu of Yangtze River Delta (YRD), and found the model performance was better for the latter inventory. Once the emission inventory of all the anthropogenic sources get improved for the whole YRD region, therefore, a better model performance with high-resolution modeling (e.g., 3km) can be expected.

*4. Considering that the PM emissions from OBB are mainly $PM_{2.5}$, and the ambient $PM_{10}$ is more affected by the local road dust emissions, it is not appropriate to only use $PM_{10}$ concentration to evaluate the model performance and analyze the contribution of OBB. I think authors shall use both $PM_{10}$, $PM_{2.5}$, CO, $NO_2$, $SO_2$, OC, EC to do the model evaluation. At least $PM_{2.5}$ shall be included considering that most Chinese cities release $PM_{2.5}$ hourly concentrations since 2013. Although authors give a couple of figures in SI, this is not enough. Specifically, the correction based on the comparisons of PM10 cannot be used for all other species.*

**Response and revisions:**

We thank the reviewer's comment. We agree with the reviewer that observation of more relevant species should ideally be included in the constraining method and evaluation of OBB emissions. However, the most and the second most fire counts were found for YRD region in 2012 and 2010 from 2005 to 2015, while the concentrations of $PM_{2.5}$, CO, $NO_2$, and $SO_2$ were unavailable before 2013. The largest daily mass ratio of $PM_{2.5}$ to $PM_{10}$ could reach 91.3% in Nanjing during the OBB event of 2012 and 77.2% in Lianyungang during the event of 2014. The contribution of OBB to $PM_{10}$ estimated in this study was 37% in YRD and 55% in Anhui province during OBB period in June 2012. The OBB could thus be identified as an important source of $PM_{10}$ during the OBB event periods as well. Therefore, we used $PM_{10}$ concentration to evaluate the model performance and analyze the contribution of OBB in 2010 and 2012. Compared to $PM_{2.5}$ and $PM_{10}$, OBB was not a major source of $NO_2$ and $SO_2$, and the OC and EC concentrations were still unavailable at present as they were not considered as regulated pollutants in China. In this case, we followed the reviewer's suggestion and applied both $PM_{2.5}$ and $PM_{10}$ concentrations to evaluate the model performance and analyze the contribution of OBB in June 7-13, 2014. Similar to 2010 and 2012, the NMBs and NMEs between observed and simulated particle concentrations with constrained OBB emissions were smaller than most of those without OBB emissions or with OBB emissions based on FRP-based. **Corresponding**

**revision was shown in lines 459-475 of Page 15 and 490-498 of Page 16 in the revised manuscript.** The average contributions of OBB to $PM_{2.5}$ and $PM_{10}$ during June 7-13, 2014 were estimated at 29% and 23% for 22 cities in YRD. It again suggested that the OBB was an important source of both $PM_{2.5}$ and $PM_{10}$ during OBB event. **Corresponding revision was shown in lines 587-593 and lines 605-607 of Page 19 in the revised manuscript.**

We also admitted the limitation of constrained method, as our response to Question 1 of Reviewer 1. We agree with the reviewer that the concentrations of $PM_{2.5}$ or OC were more suitable for constraining OBB emissions. However, the data were unavailable before 2013, particularly for 2010 and 2012 with the most and the second most fire counts detected by satellite. As OBB was an important source of $PM_{10}$ as well, we had to apply $PM_{10}$ concentrations to constrain the OBB emissions. The activity level was constrained based on the comparisons between simulated and observed $PM_{10}$ concentrations, and the OBB emissions of other species were revised according to the changed activity level. The reliability of emissions for other species depended largely on the accuracy of emission factors for $PM_{10}$ and those species. Uncertainties would be introduced to the emission estimation, resulting from lack of sufficient and qualified domestic field tests on OBB emission factors. We admit this limitation in the method section, and improvement can be expected with more measurements on concentrations of multiple pollutants and local emission factors available in the future. **Corresponding revision was shown in lines 258-264 of Page 9 in the revised manuscript.**

*5. The model performance statistics for meteorological parameters shown in Table S6 and that for PM10 concentrations as shown in Table 2 shall include the benchmark of the evaluation.*

**Response and revisions:**

We thank the reviewer's comment. The benchmarks of the evaluation for meteorological parameters from Emery et al. (2001) and Jiménez et al. (2006) were added in Table S6. The meteorological parameters of this study were basically in compliance with benchmarks. **Corresponding revision was shown in lines 312-317 of Page 11 in the revised manuscript.**

As many factors would influence the model performance of chemistry transport model, no uniform benchmark was obtained for different regions. We selected the results in US (Zhang et al., 2006) as the benchmark for $PM_{2.5}$ and $PM_{10}$ concentrations, as added in Table 2. As can be found in the table, the NMBs and NMEs for most case with the constrained OBB emissions were close to those by Zhang et al. (2006). The NMEs for hourly $PM_{2.5}$ and $PM_{10}$ were slightly larger. Given the larger uncertainty in emission inventory of anthropogenic sources for China and the uncertainty in spatial and temporal distribution of OBB emissions due to satellite detection limit, we believe the model performance with the constrained OBB emissions was improved and acceptable. **Corresponding revision was shown in lines 490-498 of Page 16 in the revised manuscript.**

*6. For OBB, temporal allocation is very important. It is good to see the monthly variations of fire occurrence in Figure 1. However, the authors only give information for year 2010 and 2012, I wonder if the authors can provide such information for other years.*

**Response and revisions:**

We thank the reviewer's suggestion and provide the information for other years (2005-2015) in Figure S3 in supplement.

*7. Figure 2 shall give the name of each city in the YRD. Otherwise it is difficult for readers to understand when author talk about Lianyungang, Fuyang, Shanghai, Suzhou, Wuxi, Changzhou, etc.*

**Response and revisions:**

We thank the reviewer's suggestion and provide the name of each city in the YRD in Figure 2.

*8. The color in Figure 4 is very difficult to read.*

**Response and revisions:**

We thank the reviewer's reminder. We applied thicker lines and changed the colors to make the figure easier to read.

[revised manuscript text omitted]
 of Zhang (2006), and the NME of hourly $PM_{2.5}$ and $PM_{10}$ with constrained OBB emissions were larger than those of Zhang (2006). Considering that the accuracy of emissions of other sources for US might be higher than that used in this study and there were uncertainties in the temporal and spatial distribution of OBB emissions derived from satellite due to insufficient satellite detection capability, the model performance with constrained OBB emissions was acceptable. However, the NMB and NME of hourly $PM_{2.5}$ and $PM_{10}$ without OBB emissions and with FRP-based emissions were higher than those of Zhang (2006).~~

[revised manuscript text omitted]


**Figure 1.**

[Figure]

(a1)    (a2)

(b1a1)    (b2a2)

(c1b1)    (c2b2)

(d1c1)    (d2c2)

**Figure 2.**

[Figure]

[Figure]

**Figure 3.**

[Figure]

**Figure 4.**

[Figure]

[Figure]

**Figure 5.**

[Figure]

[Figure]

**Figure 6.**

[Figure]

**Figure 7.**

[Figure]

**Figure 8.**

[Figure]

**Figure 9.**

[Figure]

**Figure 10.**

[Figure]

**Figure 11.**

[Figure]

---

## Referee Report (RR1)

The introduction, description of methods and analysis of data have been documented well and the results are presented in the lucid manner. Also the results showed consistent with those obtained by several other earlier studies. The quantitative estimates of OBB emissions would be helpful to study the fire impacts on regional and local air quality and therefore helpful in the policy-making in the future. But there are some flaws in this study. I recommend for publication but after substantial revision with the considerations provided below and proof-reading to strengthen the paper.

(1) Three methods, namely traditional bottom-up, fire radiative power (FRP)-based, and constraining, were used to estimate the OBB emissions. However, it's quite boring because the author does not put much insight into these methods, but simple quantify the OBB emissions with them. Actually, the bottom-up and FRP-based had been widely applied in the estimations of global or regional OBB emissions. The highlight of this study is the constraining method. This study should be emphasis on reporting the constraining method and describing its advantages relative to other methods.

(2) The spatial resolution of OBB emission inventories using three methods are also compared. So, what's the allocation factor (cropland or population?) of bottom-up-based OBB emission inventory in this study?

(3) The FRP data may miss amount of fire points because of the limitation of satellite overpass periods, leading to the underestimation of OBB emissions. The author should consider it in calculating the uncertainty of OBB emissions.

(4) compares model output using different inventories with an observational dataset. While interesting. I am interested in why the simulated PM10 level with Traditional_OBB input is significant higher than with FRP_OBB and Constrained_OBB inputs in Lianyungang, Fuyang, Bozhou and Bengbu, while no difference in Hefei and Chuzhou. In addition, more air pollutants, such as CO and $PM_{2.5}$ should be compared because OBB emissions is not the major contributor to $PM_{10}$.

(5) Specify the gird resolution in Figure 7.

---

## Author Response (AR2)

**Main revisions and response to reviewers' comments**

Manuscript No.: acp-2018-701

Title: Quantification and evaluation of atmospheric pollutant emissions from open biomass burning with multiple methods: A case study for Yangtze River Delta region, China

Authors: Yang Yang, Yu Zhao

We thank very much for the valuable comments and suggestions from the reviewer, which help us improve our manuscript. The comments were carefully considered and revisions have been made in response to suggestions. Following is our point-by-point responses to the comments and corresponding revisions.

**Reviewer #3**

*General comments: The introduction, description of methods and analysis of data have been documented well and the results are presented in the lucid manner. Also the results showed consistent with those obtained by several other earlier studies. The quantitative estimates of OBB emissions would be helpful to study the fire impacts on regional and local air quality and therefore helpful in the policy-making in the future. But there are some flaws in this study. I recommend for publication but after substantial revision with the considerations provided below and proof-reading to strengthen the paper.*

**Response and revisions:**

We appreciate the reviewer's crucial and important comments. In general, the presentation of the work has been improved, based on specific comments/suggestion from the reviewer.

*1. Three methods, namely traditional bottom-up, fire radiative power (FRP)-based, and constraining, were used to estimate the OBB emissions. However, it's quite boring because the author does not put much insight into these methods, but simple quantify the OBB emissions with them. Actually, the bottom-up and FRP-based had been widely applied in the estimations of global or regional OBB emissions. The highlight of this study is the constraining method. This study should be emphasis on reporting the constraining method and describing its advantages relative to other methods.*

**Response and revisions:**

We thank the reviewer's comment. Different methods existed in estimating the emissions from open biomass burning, but inventories were still of large uncertainty, and the discrepancies and underlying reasons were seldom analyzed. In this work we aimed to understand the reasons for the discrepancies between inventories with different methods, and thus described the data sources and principles of all the three methods. We took the reviewer's suggestion and discussed the advantages of constraining method. The constraining method did not rely on the activity levels (i.e., the burned biomass in the cropland) that were still of considerable uncertainty in China. The estimation in emissions of the species for which the ground observation was applied as constraint ($PM_{10}$ in this case) was less influenced by the uncertainties of emission factors compared to the other two methods. **Corresponding revision was shown in lines 228-233 of Page 8 in the revised manuscript.** To explore the advantages of constraining method, the OBB emissions based on the three methods were evaluated with chemistry transport model (CTM) and fire points derived by satellite, and the best model performance was achieved with the constrained OBB emissions applied in CTM, implying the reliability of the method. In contrast, the traditional bottom-up method failed to catch the actual inter-annual trend in emissions and the FRP-based method might underestimate the emissions due to limitation of satellite observation.

*2. The spatial resolution of OBB emission inventories using three methods are also compared. So, what's the allocation factor (cropland or population?) of bottom-up-based OBB emission inventory in this study?*

**Response and revisions:**

We thank the reviewer's comment. We expected that the fire radiative power (FRP) of agricultural fire point detected by satellite was a more appropriate allocation factor than cropland or population. Therefore the FRP of agricultural fire point was applied to determine the spatial pattern of OBB emissions based on the traditional method. We stated this in section 2.4 in the revised manuscript.

*3. The FRP data may miss amount of fire points because of the limitation of satellite overpass periods, leading to the underestimation of OBB emissions. The author should consider it in calculating the uncertainty of OBB emissions.*

**Response and revisions:**

We thank the reviewer's comment. The underestimation of OBB emissions based on FRP-based since the limitation of satellite overpass periods and detectability of satellite were considered in this study. According to Schroeder et al. (2008), the number of fire pixel could be underestimated by 300% on crop-dominant areas due to the both reasons, and the result was used to calculate the uncertainties of OBB emissions based on FRP-based method. **Corresponding revision was shown in lines 696-699 of Page 22 in the revised manuscript.**

*4. Compares model output using different inventories with an observational dataset. While interesting. I am interested in why the simulated PM10 level with Traditional_OBB input is significant higher than with FRP_OBB and Constrained_OBB inputs in Lianyungang, Fuyang, Bozhou and Bengbu, while no difference in Hefei and Chuzhou. In addition, more air pollutants, such as CO and*

*PM2.5 should be compared because OBB emissions is not the major contributor to PM10.*

**Response and revisions:**

We thank the reviewer's comment. The city of Lianyungang is located in northern Jiangsu, and the cities of Fuyang, Bozhou and Bengbu are located in northern of Anhui province. Indicated by the number of fire points, open biomass burning occurred more frequently in those two regions than the central Anhui where Hefei and Chuzhou are located. Compared to other anthropogenic activities, therefore, the OBB emissions and their contribution to elevated ambient particle concentrations were expected larger in Lianyungang, Fuyang, Bozhou, and Bengbu than those in Hefei and Chuzhou. In this work, for example, the average $PM_{10}$ concentration contributed by OBB for the cities of Lianyungang, Fuyang, Bozhou and Bengbu was estimated at 68 μg m$^{-3}$ in the OBB episode in 2010, and that of Hefei and Chuzhou was much smaller at 16 μg m$^{-3}$. Given more importance of OBB, the discrepancies in $PM_{10}$ concentrations simulated from various OBB inventories were thus significantly larger in cities of northern Anhui and Jiangsu than those in cities of central Anhui.

We agree with the reviewer that in general OBB is not the major contributor to $PM_{10}$. However, very few observation data for $PM_{2.5}$ and gaseous species were available before 2013. From limited data in Nanjing, we found that the mass fractions of $PM_{2.5}$ to $PM_{10}$ were 79% during the OBB episodes in 2012, implying that $PM_{10}$ could serve as an indicator of $PM_{2.5}$. We stated this in in lines 234-243 of Page 8 in the revised manuscript. Moreover, we used both $PM_{2.5}$ and $PM_{10}$ concentrations to evaluate the model performance and to analyze the contribution of OBB for 2014. Similar to 2010 and 2012, the NMBs and NMEs between the observed and simulated particle concentrations with constrained OBB emissions were smaller than most of those with FRP-based OBB emissions or without OBB emissions. The average contributions of OBB to $PM_{2.5}$ and $PM_{10}$ during June 7-13, 2014 were estimated at 29% and 23% for 22 cities in YRD, indicating that OBB was an important source of both $PM_{2.5}$ and $PM_{10}$. In addition, we also followed the reviewer's suggestion and applied CO concentrations to evaluate the model performance for the period. Similar to $PM_{2.5}$ and $PM_{10}$, the NMBs and NMEs between observed and simulated CO concentrations with constrained OBB emissions were smaller than those with FRP-based OBB emissions or without OBB emissions, implying the advantage of constrained OBB emissions against other inventories. **Corresponding revision was shown in line 466-474 of Page 15, lines 475-479 of Page 16 and lines 509-514 of Page 17 in the revised manuscript.**

*5. Specify the grid resolution in Figure 7.*

**Response and revisions:**

We thank the reviewer's suggestion and provide the horizontal resolution of Figure 7 (0.5°×0.5°).

**References**

[revised manuscript text omitted]